# Origin of dendrite-free lithium deposition in concentrated electrolytes

Yawei Chen[1,6], Menghao Li[2,3,6], Yue Liu[4,6], Yulin Jie[1], Wanxia Li[1], Fanyang Huang[1], Xinpeng Li[1], Zixu He[1], Xiaodi Ren [1], Yunhua Chen[5], Xianhui Meng[5], Tao Cheng [4]✉, Meng Gu [2]✉, Shuhong Jiao [1]✉ & Ruiguo Cao [1]✉

The electrolyte solvation structure and the solid-electrolyte interphase (SEI) formation are critical to dictate the morphology of lithium deposition in organic electrolytes. However, the link between the electrolyte solvation structure and SEI composition and its implications on lithium morphology evolution are poorly understood. Herein, we use a single-salt and single-solvent model electrolyte system to systematically study the correlation between the electrolyte solvation structure, SEI formation process and lithium deposition morphology. The mechanism of lithium deposition is thoroughly investigated using cryo-electron microscopy characterizations and computational simulations. It is observed that, in the high concentration electrolytes, concentrated Li⁺ and anion-dominated solvation structure initiate the uniform Li nucleation kinetically and favor the decomposition of anions rather than solvents, resulting in inorganic-rich amorphous SEI with high interface energy, which thermodynamically facilitates the formation of granular Li. On the contrary, solvent-dominated solvation structure in the low concentration electrolytes tends to exacerbate the solvolysis process, forming organic-rich mosaic SEI with low interface energy, which leads to aggregated whisker-like nucleation and growth. These results are helpful to tackle the long-standing question on the origin of lithium dendrite formation and guide the rational design of high-performance electrolytes for advanced lithium metal batteries.

Lithium (Li) metal is considered as an ultimate anode material due to its ultrahigh specific capacity (3860 mAh g⁻¹) and lowest standard electrode potential (−3.04 V)[1–3]. However, compared to traditional intercalation graphite anodes, Li metal anodes suffer from notorious dendrite growth issues and continuous side reactions during charging and discharging processes[3]. The dendritic growth of Li metal and the stripping inhomogeneity tend to form inactive Li (dead Li), leading to low Coulombic efficiency (CE) and failure of the battery, as well as

serious safety issues[4,5]. Therefore, the regulation of Li deposition morphologies, such as dendritic Li, mossy Li, granular Li, spherical Li, columnar Li, etc.[6] plays a vital role in improving the reversibility and CE of Li metal plating/stripping. However, the origin of Li dendrite growth and the rational design of Li deposition morphologies are still mysterious and elusive so far. Many descriptors are identified to be essential for modulating the Li deposition morphologies during the Li nucleation and growth process, including electrolyte solvation

[1]Hefei National Laboratory for Physical Science at Microscale, CAS Key Laboratory of Materials for Energy Conversion, Department of Materials Science and Engineering, University of Science and Technology of China, Hefei 230026, China. [2]Department of Materials Science and Engineering, Southern University of Science and Technology, Shenzhen 518055, China. [3]School of Materials Science and Engineering, Harbin Institute of Technology, Harbin 150001, China. [4]Institute of Functional Nano and Soft Materials (FUNSOM), Soochow University, Suzhou 215123, China. [5]NIO Incorporation, Shanghai 201800, China. [6]These authors contributed equally: Yawei Chen, Menghao Li, Yue Liu. ✉e-mail: tcheng@suda.edu.cn; menggu1985@hotmail.com; jiaosh@ustc.edu.cn; rgcao@ustc.edu.cn

structure[7,8], substrate effect[9], ionic diffusion[10], temperature[11,12], pressure[13], etc.

The electrolyte solvation structure is an accessible and widely used descriptor to evaluate Li metal compatibility of electrolytes. Currently, numerous advanced strategies are used to modulate the electrolyte solvation structure to stabilize both Li metal anode and high-voltage cathodes, such as high-concentration electrolytes (HCEs)[14–20], localized high-concentration electrolytes (LHCEs)[21–25], fluorinated solvents[26–32], highly sterically hindered solvents[33] and so on. Essentially, these strategies can increase the proportion of anions in the solvation sheath while decreasing the relative proportion of solvents. This anion-dominated solvation structure facilitates the formation of denser inorganic-rich interphase layer to passivate Li metal[31,34]. Solid-electrolyte interphase (SEI) formed at the electrode-electrolyte interface has been considered the most important factor for regulating the Li plating morphology. Recently, cryo-electron microscopy has characterized SEI structures at the atomic scale[35,36]. Accordingly, two distinct SEI structures are currently recognized, mosaic and multilayer structures[24,35,37]. In the mosaic structure, inorganic nanoparticles are randomly distributed in the organic matrix. While in the multilayer counterpart, the organic and inorganic layers are uniformly arranged to form an ordered layered structure. It is believed that the composition and structure of the SEI depend on the reduction reaction activity sequence of the electrolyte components on the Li metal surface[38], and the solubility and reduction stability of the interfacial decomposition products. However, no consistent conclusions have emerged to explain the intrinsic motivation for forming these SEI structures. The comparison of different SEI structures in previous studies has been carried out in different electrolyte systems or new additives[24,35,39,40], which results in not only changes in solvation structures but also differences in interfacial chemical compositions. Furthermore, a unified descriptor is lacking for the effects of SEIs with different structures and compositions on Li deposition behavior, although many debates persist including interphase uniformity[41], mechanical stability[11], ionic conductance[14], SEI swelling[40], interface energy[42], fluorinated interphases[16,17], inorganic-rich interphases[43], etc. In addition, the multi-component configurations in the electrolyte exacerbate the complexity of the interfacial chemical/electrochemical reactions[40,44], thus requiring simpler and representative electrolyte systems for investigation.

Herein, we select the commonly used lithium bis(fluorosulfonyl)imide/1,2-dimethoxyethane (LiFSI/DME) combination to explore the interaction mechanism between the solvation structure of the electrolyte, the SEI structure and the Li deposition morphology by modulating the electrolyte concentration. The SEI structures are characterized by cryo-electron microscopy, and it is found that the SEI exhibits a mosaic structure at low concentration. In contrast, the SEI formed at high concentrations has a uniform and complete amorphous structure. The combination of XPS characterization and theoretical simulation calculation shows that the fast salt decomposition dominates at high concentrations, similar to a supercooling state in the growth process of amorphous metal materials, resulting in the generation of amorphous SEI. While at low concentrations, the decomposition of DME solvent is much more vigorous, and the relatively slow reaction of salts contributes to the crystallization process of inorganic nanoparticles inside SEI. Compared with the mosaic-structured SEI containing organic matrix and inorganic crystalline particles, the inorganic-rich amorphous-structured SEI probably has higher interface energy, which can facilitate the growth of granular Li or columnar Li instead of mossy Li or Li dendrites. Our fundamental studies on the origin of dendrite/dendrite-free formation in low/high concentration electrolytes can deepen the understanding of electrolyte solvation structure design, SEI growth behavior, and their modulation on Li deposition morphology.

## Results

### Li deposition morphology

The appearances of the electrolytes of LiFSI in DME with different salt concentrations are shown in Supplementary Fig. 1, where the salt-to-solvent molar ratios range from 1:10, 1:4, 1:2, 1:1.4, 1:1 to 1:0.7 (saturate concentration, 25 °C), which are denoted as LiFSI-xDME ($x$ = 10, 4, 2, 1.4, 1, 0.7, respectively). The electrolytes with $0.7 \leq x < 2$ can be considered high concentration electrolytes (>3 M, HCEs). All the solutions are homogeneous transparent liquids at room temperature (Supplementary Fig. 1). Changing the salt concentration is the most direct and effective way to modulate the solvated structure of the electrolyte. Supplementary Fig. 2 captures a significant and representative region of 690–820 cm⁻¹ corresponding to the S–N–S bending vibration of FSI⁻ anions. The bands at 718 cm⁻¹ arise from the free FSI⁻ anion, and the band at ~730 cm⁻¹ belongs to single contact-ion pair (CIP), in which one FSI⁻ anion is only associated with one Li⁺. The band at ~750 cm⁻¹ arises from an aggregate ion pair (AGP), in which one FSI⁻ anion interacts with two or more Li⁺. As the concentration of LiFSI increases, free FSI⁻ anions gradually decrease until disappearing in the LiFSI-1.4DME electrolyte and more concentrated electrolytes. In addition, the percentage of AGP reaches 99.6% in the LiFSI-0.7DME electrolyte. Overall, the solvated electrolyte structure has a more free and CIP structure under low-concentration conditions, while AGP and CIP are dominant in high concentration counterparts.

To investigate the Li nuclei and growth morphology evolution during Li deposition, as shown in Fig. 1 and Supplementary Fig. 3, fixed capacities (0.02–1 mAh cm⁻²) of lithium metal were deposited on the Cu electrodes at the current density of 0.5 mA cm⁻². The sampling points of SEM images are randomly selected in the center of the electrode, avoiding the uneven deposition areas on the electrode edges. At the low capacity of 0.02 mAh cm⁻², the deposited Li metal exhibits an aggregate whisker-like appearance at low concentrations (Fig. 1a and Supplementary Fig. 3a–c). Especially in the LiFSI-10DME electrolyte, the whiskers are scattered in large aggregates, which indicates that its Li nucleation sites are few and unevenly distributed. With the increase in concentration, in the high-concentration LiFSI-1.4DME electrolyte, Li nucleation and initial growth show a dense granular morphology without whisker-like Li. Surprisingly, uniform disc-like and spherical morphologies were obtained in the LiFSI-1DME electrolyte. Under the saturated concentration (LiFSI-0.7DME), smaller spherical Li nuclei with a small amount of Li whisker were observed. The Li whisker formation could be related to the limited diffusion of Li⁺ caused by the high viscosity of the electrolyte. Overall, as the salt concentration increases, the morphology of electrodeposited Li nuclei gradually changes from whisker aggregates to granular and finally into a regular spherical shape. Correspondingly, the size of Li nuclei gradually decreases (Fig. 1a), accompanied by an increase in the number and uniformity of Li nuclei as the concentration increases (Supplementary Fig. 3a–f). At a higher capacity of 0.1 mAh cm⁻², the salt concentration effect causes the deposited Li to exhibit two main morphological features (Fig. 1b), the curved whisker-like Li at low concentration and the dense granular Li at high concentration. Similarly, the Li nuclei density and uniformity gradually increase as the salt concentration increases, accompanied by the larger coverage areas of Li deposits on the Cu current collector. Figure 1c, d show SEM images of Li deposition morphology at 1 mAh cm⁻² capacity in LiFSI-xDME electrolytes. Similar to the nucleation stage, more extensive substrate coverage and more uniform Li deposition are achieved with increasing concentrations. Especially under high concentration conditions (LiFSI-xDME, $x \leq 1.4$), the deposited Li displays a relatively flat and dense morphology (Fig. 1d, e), and all showed a uniform large granular structure (Fig. 1c). However, the island-like Li aggregation distributions under lower concentration conditions (LiFSI-xDME, $x \geq 2$) lead to incomplete coverage of the Cu current collector with uneven Li deposition (Fig. 1d, e). A small amount of whisker-like Li can also be

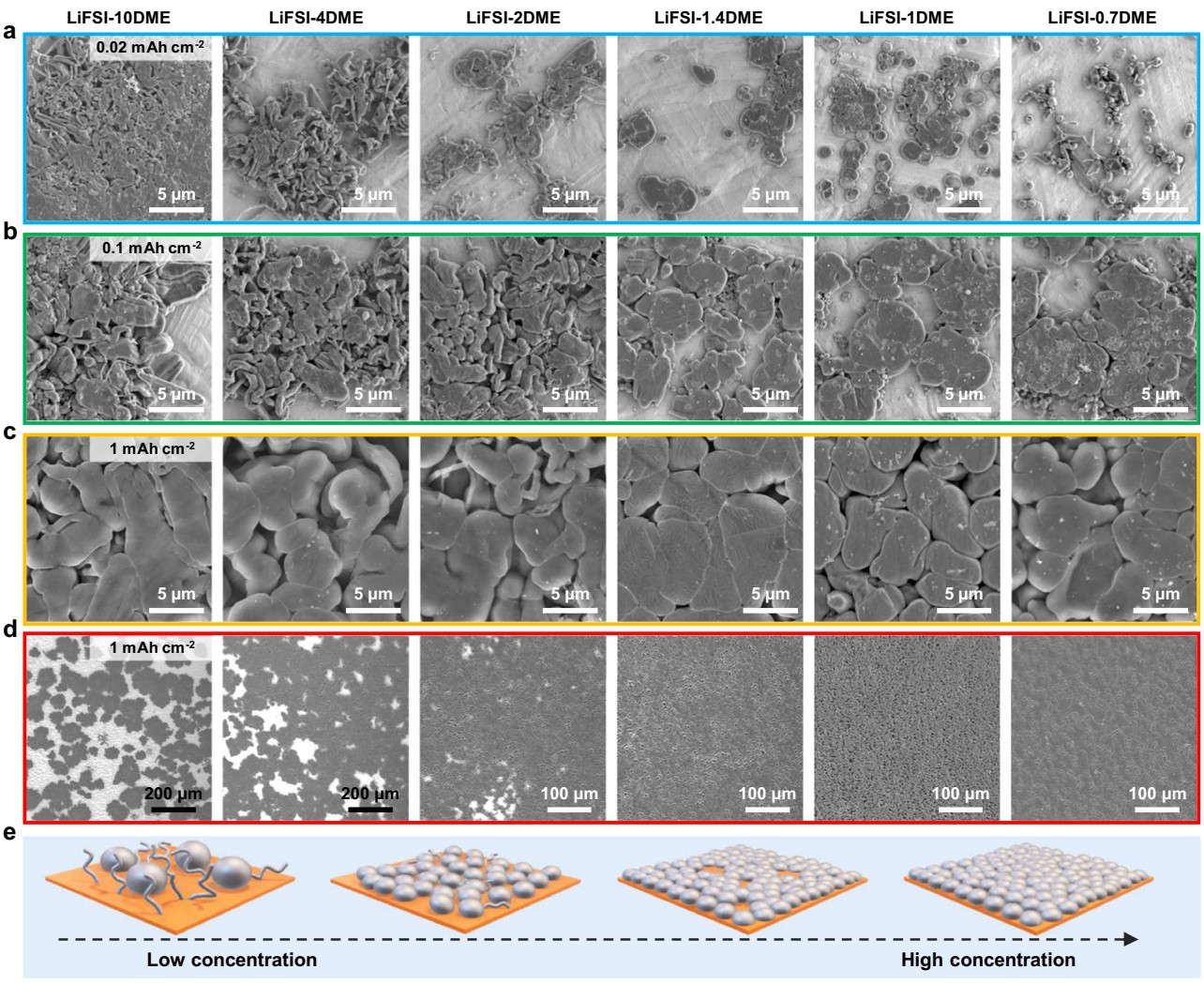

**Fig. 1 | Li deposition morphology evolution in electrolytes of different concentrations. a–d** SEM images of Li deposited on Cu foils in LiFSI-xDME electrolytes ($x$ = 10, 4, 2, 1.4, 1, 0.7, respectively) at **a** 0.02 mAh cm⁻², **b** 0.1 mAh cm⁻², **c, d** 1 mAh cm⁻², and the current density is 0.5 mA cm⁻². **e** Schematic of the concentration effect of Li electrodeposition on Cu.

observed in the gaps of granular Li deposits (Fig. 1c, e). Combining with these results, the electrodeposited Li on Cu in the low-concentration electrolytes exhibits a whisker-like morphology with a high specific surface area accompanied by inhomogeneous nucleation and growth. While under high concentration conditions, uniform spherical and granular Li nucleation and growth enable the deposited Li with a denser structure and smaller specific surface area, which reduces the reactivity of the Li metal/electrolyte interface.

Such different features of Li deposition morphology lead to differences in the compatibility of electrolytes with Li metal. The CEs of Li metal in electrolytes were determined by the ratios of stripping capacity and plating capacity on Cu electrodes in Li/Cu half cells (Supplementary Fig. 4). The Li CEs exhibit a volcano plot trend with increasing concentration (Supplementary Fig. 5). The LiFSI-10DME electrolyte exhibits the lowest CE (89.64%), and the CEs gradually increase with increasing concentration until the highest CE of 99.20% is achieved in the LiFSI-1.4DME electrolyte. After that, higher concentrations instead show lower CEs, which are consistent with previous findings on HCEs[14]. However, the stability of CEs improves with the increase in electrolyte concentration. Under lower concentration conditions (LiFSI-xDME, $x \geq 2$), the unstable fluctuation of CEs is significant in the later cycling stage. At the same time, it shows better cycling stability with occasional fluctuations under high concentration

conditions (>1000 cycles). Furthermore, the long-term stability of Li/Li symmetric cells shown in Supplementary Fig. 6 displays similar results (>4000 h with only a slight increase in overpotential). The exchange current densities of different electrolytes were calculated according to the Tafel plots for Li plating/stripping (Supplementary Fig. 7). High-concentration electrolytes possess lower exchange current densities (0.15–0.22 mA cm⁻²) compared to low-concentration counterparts (0.22–6.49 mA cm⁻²), and the LiFSI-1.4DME electrolyte shows the lowest exchange current density, which corroborates with its improved CE in the Li/Cu half-cell. Previous studies have shown that lower exchange current density benefits the formation of larger radius Li nuclei[45], which is also consistent with our results. The deposited granular Li in high concentration electrolytes has a denser and lower surface area structure, which facilitates Li stripping (Supplementary Fig. 8) and reduces interfacial side reactions, resulting in higher CEs. Moreover, as the electrolyte concentration increases, the thickness of the passivation layer formed on the Cu electrode surface gradually decreases during the long cycle (30–8.6 μm, Supplementary Fig. 9).

## Interphasial structure and chemistry
Cryogenic transmission electron microscopy (cryo-TEM) was implemented to characterize the structure and composition of SEI formed at the Li-electrolyte interface. Three representative electrolytes (LiFSI-

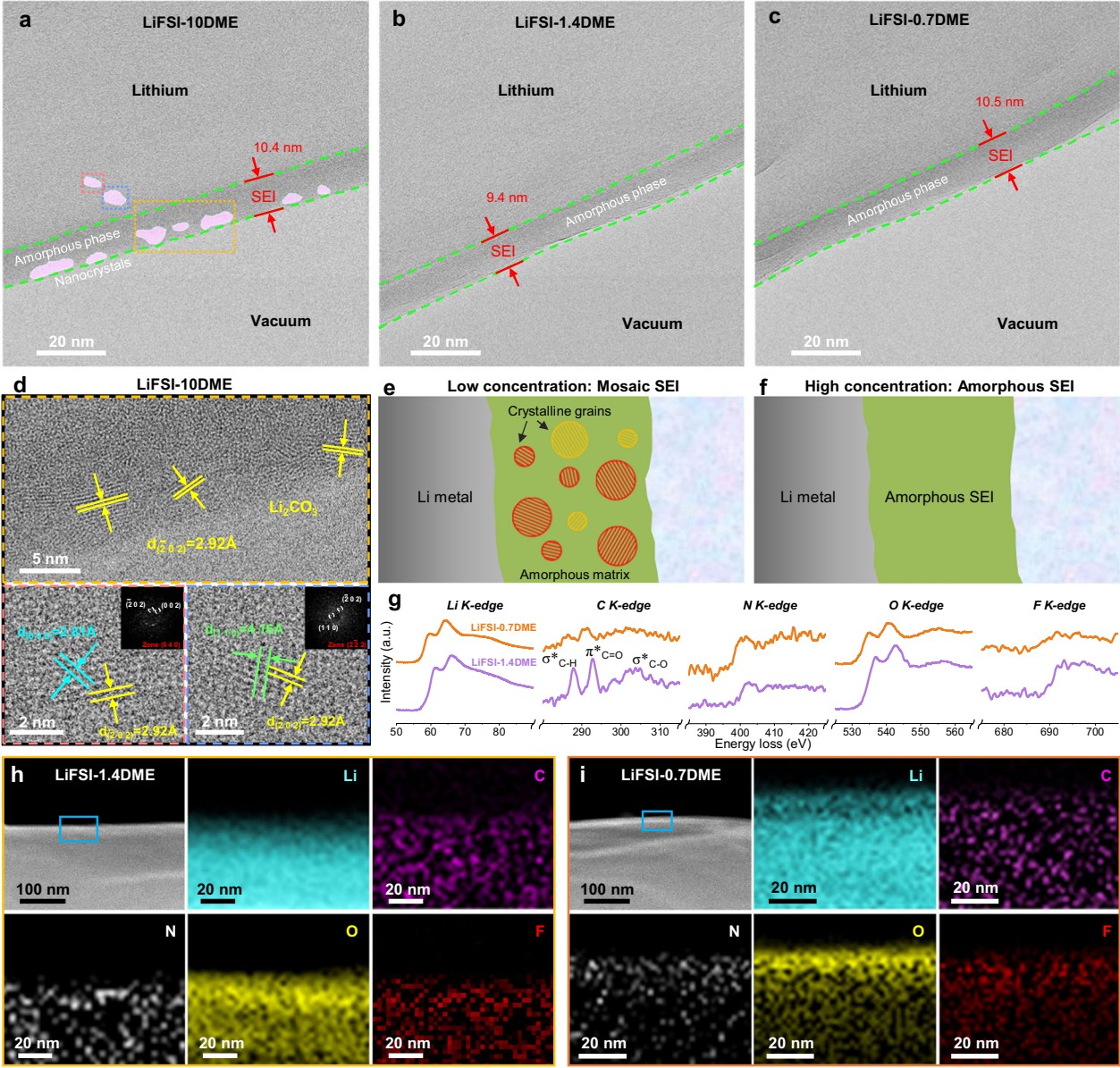

**Fig. 2 | Cryo-EM characterizations of SEI in electrolytes of various concentrations. a**–**c** Cryo-TEM images of SEI on Li metal in LiFSI-xDME electrolytes (*x* = 10, 1.4, 0.7, respectively). **d** HRTEM images and corresponding fast Fourier transform (FFT) of SEI growth on Li in the LiFSI-10DME electrolyte. **e, f** Schematic illustrating of SEI structures in low and high concentrations. **g** Cryo-TEM/EELS analysis of SEI on deposited Li in LiFSI-xDME electrolytes (x = 1.4 and 0.7). **h, i** Cryo-STEM ADF images and EELS mapping images of SEI on Li metal in the **h** LiFSI-1.4DME and **i** LiFSI-0.7DME electrolytes.

10DME, LiFSI-1.4DME, and LiFSI-0.7DME) were selected for the interface research. Supplementary Fig. 10 shows the low-magnification cryo-TEM images of deposited Li on TEM Cu meshes in these three electrolytes. Inhomogeneous aggregated Li deposition on the Cu mesh can be observed in the LiFSI-10DME electrolyte (Supplementary Fig. 10a). The accumulation of Li whiskers inside the deposited Li can be identified by cryo-TEM image (Supplementary Fig. 10d). In the high/ ultrahigh concentration electrolytes (LiFSI-1.4DME and LiFSI-0.7DME), a more uniform distribution of Li deposits on the Cu collector (Supplementary Fig. 10b, c) and the deposited Li shows flake morphology with small fractions of Li whiskers (Supplementary Fig. 10e, f), which can be attributed to the pressure-free Li deposition at the edge of the Cu mesh. High-resolution cryo-TEM images at an atomic scale are further carried out to determine the SEI structure and composition, as shown in Fig. 2. The green dashed line in Fig. 2a labels the boundaries of the SEI formed at the Li-electrolyte interface in the LiFSI-10DME electrolyte, and the SEI layer has an average thickness of ~10.4 nm.

Small crystalline nanoparticles (diameter 3~6 nm, marked by purple) with clear lattice fringes disperse randomly in an amorphous matrix. The zoom-in images shown in Fig. 2d indicate these crystalline nanoparticles as inorganic components of the SEI, such as $Li_2CO_3$, by matching lattice spacings. Such a SEI structure with a heterogeneous dispersion of inorganic nanoparticles in the amorphous matrix is generally defined as the mosaic SEI structure (Fig. 2e). However, a completely different SEI structure can be observed in the LiFSI-1.4DME electrolyte with a high salt concentration. A homogeneous monolayer amorphous SEI without any crystalline grains (Fig. 2f) is formed on the Li surface in the LiFSI-1.4DME electrolyte. The consistent SEI thickness of ~9.4 nm (Fig. 2b) is slightly lower than that in the LiFSI-10DME electrolyte. Similarly, for the ultrahigh concentration counterpart (LiFSI-0.7DME), an amorphous SEI layer with a higher thickness of ~10.5 nm is presented in Fig. 2c. To analyze the composition of amorphous components of SEI in the high/ultrahigh concentration electrolytes, cryogenic electron energy loss spectroscopy (cryo-EELS) is

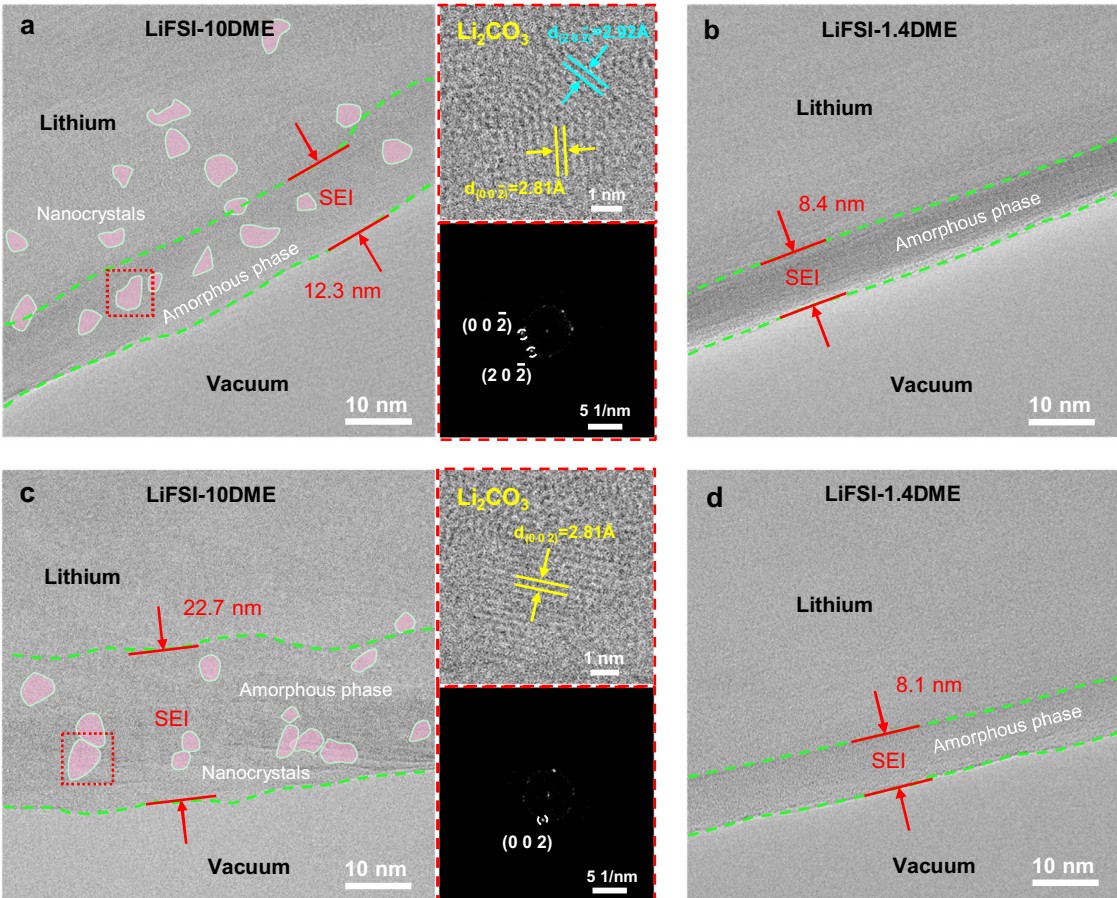

**Fig. 3 | Microstructure of the SEI formed at a low current density and after multiple cycles.** **a**, **b** Cryo-TEM images of SEI formed on the initial Li deposition in **a** LiFSI-10DME and **b** LiFSI-1.4DME electrolytes at 0.1 mA cm⁻² current density. **c**, **d** Cryo-TEM images of SEI formed on the freshly deposited Li after 5 cycles in **c** LiFSI-10DME and **d** LiFSI-1.4DME electrolytes. The right sides of **a** and **c** show HRTEM images and corresponding FFT images of the selected regions.

conducted. Figure 2g illustrates the EELS spectra of Li, C, N, O, and F K-edges of the SEI formed in the LiFSI-1.4DME and LiFSI-0.7DME electrolytes. The EELS Li K edge and O K edge of the SEI formed in these two electrolytes both display signature peaks of $Li_2O$ and $Li_2CO_3$. The EELS N K edge and F K edge can be attributed to $Li_xN$, LiF and residual LiFSI salts in the SEI. Notably, the F K edge peak intensity of the SEI in the LiFSI-1.4DME electrolyte is stronger than that in the LiFSI-0.7DME electrolyte, which reveals the higher LiF content in the SEI formed in the LiFSI-1.4DME electrolyte, excluding the interference of residual LiFSI salts. The C K edge EELS fine structures show similar features of C−H (σ*), C=O (π*) and C−O (σ*) in the SEI formed in both electrolytes, but the SEI formed in the LiFSI-1.4DME electrolyte has much sharper and stronger signals possibly due to a higher content of amorphous inorganic composition or a higher mass density of the as-formed SEI. This result indicates minimal solvolysis (DME) in the ultra-concentrated system (LiFSI-0.7DME), which is consistent with its low solvent content. Cryo-scanning TEM (cryo-STEM) EELS mapping of the SEI on the Li surface in both electrolytes is further carried out to identify the spatial distribution of the main elements (Li, C, N, O, and F). Figure 2h, i shows that these elements are uniformly distributed in the amorphous SEI matrix without obvious segregation. Such a homogeneous amorphous SEI can achieve more uniform Li⁺ ion diffusion than the heterogeneous mosaic SEI. In addition, monolithic amorphous SEI can maintain a more stable structure with better mechanical properties and flexibility during Li growth.

We also characterized the Li metal deposition morphology and SEI microstructure under a lower current density condition (0.1 mA cm⁻²) by cryo-TEM. We choose the representative LiFSI-10DME

and LiFSI-1.4DME electrolytes for our study, and the Li deposition capacity is still fixed at 0.5 mAh cm⁻². As shown in Supplementary Fig. 11a−c, the low-magnification cryo-TEM images of deposited Li with the LiFSI-10DME electrolyte display a denser morphology at a lower current density. We also note that the SEI still maintains the mosaic structure (Fig. 3a, Supplementary Fig. 11d−f). In the high-concentration LiFSI-1.4DME electrolyte, the deposited Li displays a dendrite-free and granular Li morphology (Supplementary Fig. 12a−c). Interestingly, even at the low current density of 0.1 mA cm⁻², the SEI still exhibits an amorphous structure (Fig. 3b, Supplementary Fig. 12d−f). In addition, the average thickness of the SEI in LiFSI-1.4DME (8.0 nm) is significantly lower than that of LiFSI-10DME (12.2 nm), which further demonstrates the effective passivation of the interface in highly concentrated electrolytes.

To evaluate the robustness of the SEI with different structures, we further carried out the cryo-TEM observation of the SEI morphology after Li stripping. When using the LiFSI-10DME electrolyte, the stripped Li metal exhibits obvious inhomogeneity with a large amount of dead Li (Supplementary Fig. 13a−d). Many dendrites are partially stripped, and notched structures appear at the kinks of the dendrites (Supplementary Fig. 13c, d). As clearly shown by the enlarged cryo-TEM image in Supplementary Fig. 13e, the crack of SEI appears at the notched region. By contrast, under the high concentration condition, deflated empty SEI husks display a complete and unbroken morphology without visible dead Li (Supplementary Fig. 14), which corresponds to its high CE.

After the 5th cycle of Li plating and stripping, the newly deposited Li in the LiFSI-10DME electrolyte exhibits severe Li dendrite

morphology (Supplementary Fig. 15a–c) with a large number of deflated SEI husks (Supplementary Fig. 15a, d–f). The existence of deflated SEI husks indicates that the subsequent Li metal is difficult to deposit into the SEI shells. More electrolyte is consumed to form the new SEI, resulting in the accumulation of SEI husks after multiple cycles, which in turn leads to a sharp increase in the thickness of the Li metal anode (Supplementary Fig. 9m). More importantly, the SEI formed on the deposited Li metal still exhibits a mosaic structure after multiple cycles, and the thickness distribution of SEI also shows obvious inhomogeneity (9.5, 21.1, 22.7 nm, Fig. 3c and Supplementary Fig. 15g–i). This indicates that two or more layers of SEI may accumulate at the Li interface during cycling, further confirming the interphasial instability in low-concentration electrolytes. In sharp contrast, in the LiFSI-1.4DME electrolyte, the deposited Li after 5 cycles display granular and columnar morphology with few dendrites. Interestingly, the deflated SEI husks are absent (Supplementary Fig. 16a–f), which indicates that Li metal can re-grow inside the SEI void shells. Surprisingly, even after multiple cycles, the amorphous structure SEI can still be kept, and a dense SEI with an average thickness of 8.4 nm is formed (Fig. 3d, Supplementary Fig. 16g–i). These results further indicate that the amorphous SEI formed in high concentration electrolytes endows better stability and Li metal compatibility than the mosaic SEI in low-concentration electrolytes.

Quantitative nanomechanical mapping (QNM) based on atomic force microscopy (AFM) was used to quantify the SEI modulus and its distribution on the Li surface (Supplementary Figs. 17 and 18). As shown in Supplementary Fig. 18, the SEI layer formed in the LiFSI-10DME electrolyte possesses a low average modulus of ~0.73 Gpa. As the salt concentration of the electrolyte increases, the average modulus of the formed SEI shows a gradually increasing trend. In high concentration electrolytes (LiFSI-$x$DME, $x \leq 1.4$), the average modulus of SEI is higher than 1.8 Gpa, significantly higher than the low-concentration systems (LiFSI-$x$DME, $x \geq 2$). The SEI modulus difference at different concentrations may be related to the content of inorganic components inside SEI. Theoretically, the higher salt concentration, the stronger reactivity of the anion, the more inorganic components produced by anion decomposition, and the higher SEI modulus.

Depth-profiling X-ray photoelectron spectroscopy (XPS) was further carried out to reveal the chemical components of the SEI formed on electrodeposited Li metal surface in LiFSI-$x$DME electrolytes ($x = 10$, 1.4, 0.7, respectively) (Fig. 4a-c and Supplementary Fig. 19). The SEI components are mainly $Li_2O$, $Li_2CO_3$, LiF, $Li_2S$, organic C–O group and so on, and the relative contents of these components are shown in Fig. 4d. Before $Ar^+$ ion sputtering, the outer layer of SEI is dominated by $Li_2CO_3$, which is consistent with the cryo-TEM result that $Li_2CO_3$ nanocrystals are distributed in the outer layer of SEI in Fig. 2a. After 2-min sputtering, the content of $Li_2CO_3$ is greatly reduced, and $Li_2O$ instead becomes the main component of the inner layer of the SEI. In addition, inorganic LiF and $Li_2S$ contents increase significantly, while the organic –C–O and –O–C=O content decrease drastically, which indicates that the organic components are mainly distributed in the outer layer of the SEI. The LiF content in the SEI formed in the LiFSI-1.4DME electrolyte is higher than that in the other two electrolytes, consistent with the EELS F K edge in Fig. 2g. More importantly, the peaks assigned to $Li_2C_2$ (or Li–C) in the Li 1$s$ (-52.2 eV) and the C 1$s$ spectra (-282.3 eV) gradually weaken with increasing salt concentration. $Li_2C_2$ has a strong Raman selectivity, so its existence can be further verified by Raman spectroscopy. Figure 5a displays representative Raman spectra of the surfaces of Li metal deposited in different concentration electrolytes, in which the peak at 1848 cm$^{-1}$ can be assigned as the C-C stretching mode of $Li_2C_2$. A strong $Li_2C_2$ signal was detected on the Li surface in the LiFSI-10DME electrolyte, while this signal became extremely weak at high concentration conditions and even disappeared in the LiFSI-0.7DME electrolyte. The relative content of $Li_2C_2$ in SEI can be represented by the peak area of the signal at

1848 cm$^{-1}$. On this basis, 2D mapping of $Li_2C_2$ component distributions on the Li surface can be obtained (Fig. 5 b–d). In the LiFSI-10DME electrolyte, the formed SEI contains a large amount of $Li_2C_2$ with uneven distribution (Fig. 5b). Under high/ultrahigh concentration conditions, the $Li_2C_2$ formed in SEI is relatively uniform, and the content is greatly reduced (Fig. 5c, d). $Li_2C_2$ component gradually decreases with the increase of salt concentration, indicating that the solvent's decomposition can be suppressed by adjusting the salt concentration in the electrolyte, thereby forming a denser and more stable SEI. In addition, $Li_2C_2$ has strong electrical conductivity, and its large distribution in the SEI may lead to the deterioration of the electrical insulation of the SEI, which in turn causes the continuous decomposition of the electrolyte and aggravates the instability of the interface.

## Interfacial reaction from simulations

Hybrid ab initio and reactive molecular dynamics (HAIR-MD) simulations were carried out to elucidate interphase reactions and the SEI formation process on Li metal electrodes. Figure 6a shows the initial structures of the Li metal/electrolyte interphase with LiFSI: DME ratios of 10, 1.4, and 0.7. After 2.8 ns HAIR-MD simulation, the simulated interphase structures are significantly different in these three electrolytes. As shown in Fig. 6b, at the low concentration (LiFSI-10DME), the FSI⁻ anion decomposes incompletely, producing a small amount of $Li_2O$, LiF, and $Li_3N$ but with no $Li_2S$, a product from FSI⁻ deep decomposition. Moreover, the produced $Li_2O$ aggregates into clusters, consistent with the mosaic SEI structure observed experimentally (Fig. 2a). These are the inorganics components revealed from the HAIR-MD simulation. The reduction of DME solvent occurs at around 115.6 ps in the LiFSI-10DME electrolyte. As shown in Fig. 6c, the decomposition starts from the formation of a Li⁺-DME complex. Li⁺ cation interacts strongly with the lone pairs of O in DME, weakening the C–O bond. The C–O breaking coupled with two electron transfers leads to the formation of one $C_2H_4$ and two $CH_3OLi$. In the 2.8 ns HAIR simulation, the deep products from DME, such as LiH, HCHO, and $Li_2C_2$, also appear. The statistics of these species, together with anion reduction products, are shown in Fig. 6d. The presence of these products, such as $Li_2C_2$, is consistent with the experiment XPS and Raman spectra (Figs. 4a and 5a, b). These simulation results confirm that the DME solvent undergoes reduction at low-concentration electrolyte. Instead, no DME decomposition is observed in the high-concentration electrolyte. As shown in Fig. 6b, d, only FSI⁻ anions decompose in 2.8 ns HAIR simulation, producing $Li_2O$, LiF, $Li_2S$, $Li_3N$ and $LiN_xS_yO_z$ in LiFSI-1.4DME and LiFSI-0.7DME electrolytes. These inorganic products accumulate into the inorganic inner layer (IIL) of SEI, which develops into the the amorphous structure as observed experimentally (Fig. 2b, c). The advantages of suppressing DME decomposition are at least two-fold. First, the IIL of SEI is more robust and electron-resisted, which provides better protections to Li anode. Second, the reduced gas productions, such as $C_2H_4$, mitigates the structural damage of the SEI due to gas release.

The electrochemical reactivities of the FSI⁻ anion and DME solvent can be distinguished by the cyclic voltammetry (CV) tests at a scan rate of 1 mV s$^{-1}$ on Cu electrodes before Li metal deposition (2.7-0 V vs. Li/Li⁺). As shown in Supplementary Fig. 20, the reduction peaks at -1.7 V can be attributed to the reductive decomposition process of FSI⁻ anion[46], and the reductive decomposition process of the DME solvent is shown as a subsequent sloping curve. In the lower concentration electrolytes (LiFSI-10DME and LiFSI-4DME), the -1.7 V reduction peak is absent and indistinguishable. In the LiFSI-2DME electrolyte, the CV reduction curve has a slight fluctuation at -1.7 V. Whereas under high concentration conditions (LiFSI-$x$DME, $x \leq 1.4$), the peak intensities gradually increase with the increase in concentrations (Supplementary Figs. 20 and 21). In addition, the current densities for solvolysis at high concentrations is lower than that at low concentrations. These results directly indicate that anions are

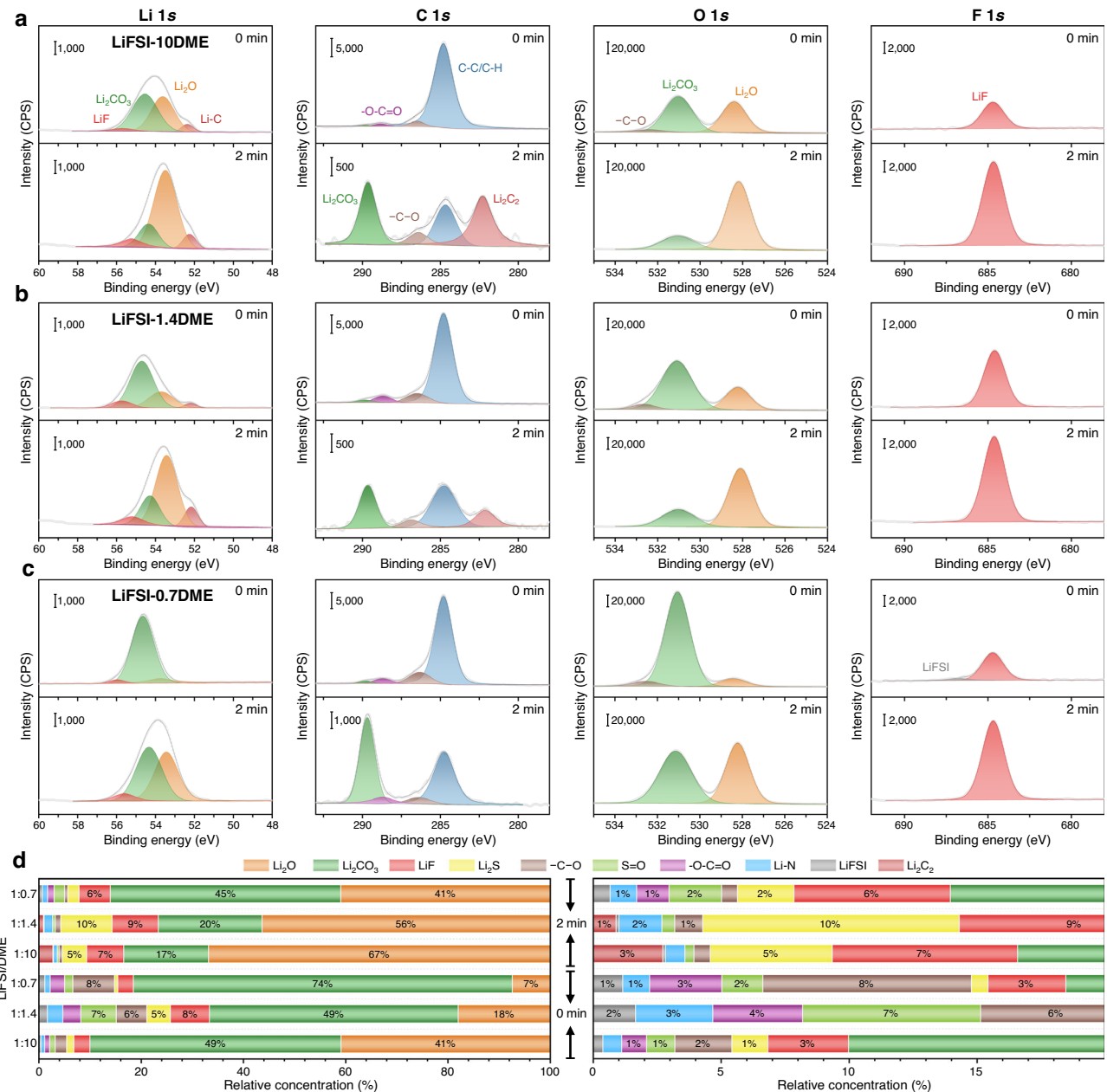

**Fig. 4 | SEI composition obtained by XPS measurement.** Li 1*s*, C 1*s*, O 1*s*, and F 1*s* spectra of SEI on Li metal at different sputtering times (0 min and 2 min) in the **a** LiFSI-10DME, **b** LiFSI-1.4DME and **c** LiFSI-0.7DME electrolytes. **d** Quantified results of the relative concentrations of decomposition species composition ratios of the SEI.

more easily decomposed with increasing concentration and the solvolysis is partially suppressed, in line with the cryo-TEM, XPS, Raman mapping and theoretical simulation results.

Figure 7 schematically illustrates the effects of the solvated electrolyte structure and SEI structure on Li metal deposition morphology adjusted by the salt electrolyte concentrations. In the low-concentration electrolyte, diluted Li$^+$ concentration lowers Li metal nucleation kinetics, resulting in heterogeneous nucleation and agglomerative growth of Li metal. Simultaneously, diluted FSI$^-$ concentration and the free/CIP structure of FSI$^-$ anions kinetically and thermodynamically reduce their reactivity at the Li metal interface. Such a slower salt decomposition process can facilitate the inoculating crystals of the inorganic components produced by the salt decomposition at the interface, thus yielding a mosaic SEI structure. More DME solvent molecules enter the solvation sheath and easily react with Li metal to generate organic-rich SEI. This organic-rich mosaic SEI

possesses a lower modulus and toughness, which is insufficient to inhibit the growth of lithium dendrites. Organic components and inorganic crystalline particles have lower interface energies than amorphous inorganics. Thermodynamically, this low-interface-energy SEI may lead to the growth of Li metal with a larger specific surface area, that is, whisker growth.

In the high concentration electrolytes, the lowered Li nucleation barrier contributes to uniform and abundant Li nucleation. Multiple Li$^+$ coordination by the FSI$^-$ anions in solvation sheath (CIP/AGP) can cause the location of the lowest unoccupied molecular orbital (LUMO) of electrolytes to shift to the anions instead of the solvent, predominant reductive decomposition of the salts on the surface of Li metal to form an anion-derived inorganic-rich interphase film (Li$_2$O, LiF, and Li$_2$CO$_3$). The fast interfacial reaction kinetics of salt decomposition leads to the rapid generation of inorganic products in the SEI without sufficient crystallization process, resulting in inorganic-rich

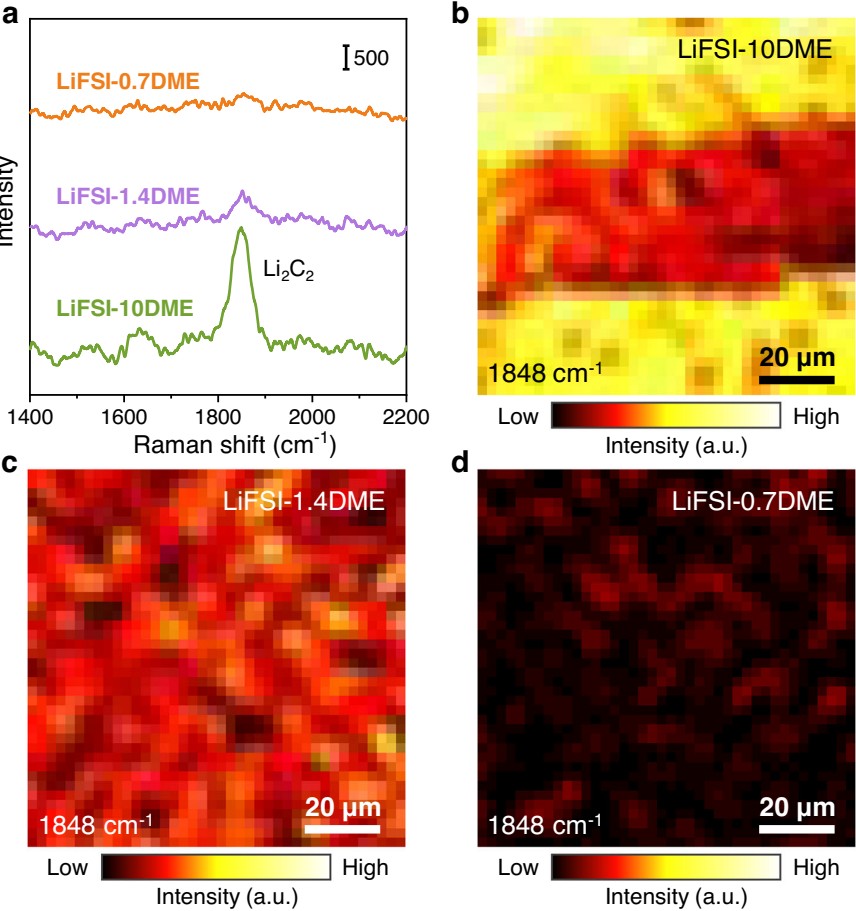

**Fig. 5 | Raman characterization of the Li₂C₂ component in the SEI on the deposited Li metal. a** Representative Raman spectra of the SEI formed in LiFSI-xDME electrolytes (x = 10, 1.4, 0.7, respectively) at a wavenumber range of 1400-2200 cm⁻¹. **b–d** The corresponding 2D mapping of Raman signal intensities at 1848 cm⁻¹.

amorphous SEI. The formation process of this amorphous SEI is similar to the supercooling process of amorphous metal growth. Such inorganic-rich amorphous SEI exhibits higher modulus, toughness, and, more importantly, higher interface energy. The SEI with high interface energy facilitates the growth of granular Li with a low specific surface area.

## Discussion

In summary, by examining the differences in the electrolyte solvation structure, the Li deposition morphology and the SEI structure in the single-salt and single-solvent LiFSI/DME electrolyte, we have found that the composition and structure of SEI are critical for regulating Li metal deposition behavior. Experimental and computational evidence indicates that a rapid and complete anion decomposition process is implemented at the Li metal interface in high concentration electrolytes with an anion-dominated solvation structure. Inorganic components in SEI undergo an amorphization process due to this fast reaction and finally form an inorganic-rich amorphous SEI. Such amorphous SEI poses high specific interface energy, so the electrodeposited Li metal exhibits a granular morphology to reduce the total interface energy. In low-concentration electrolytes, the decomposition of the solvent and the slow reduction reaction of anions promote the partial crystallization of inorganic components in the organic matrix to form mosaic SEI. Such organic-rich mosaic SEI with lower specific interface energy cannot effectively inhibit dendrite growth thermodynamically, resulting in the formation of whisker Li with a high surface area. We believe that our fundamental understandings can

potentially help design SEI structure and composition through solvation structure control and further understand the regulation of SEI structure on Li deposition morphology.

## Methods

### Materials

DME (99.95%) and LiFSI (99.8%) were purchased from Guotai Huarong New Chemical Materials Co., Ltd. LiFSI was dried by vacuum at 95 °C for 12 h before use. DME is dewatered by activated 4 Å molecular sieves for 48 h before use. Li foils (450 μm in thickness, 15.6 mm in diameter) was purchased from China Energy Lithium Co., Ltd. Cu foil (25 μm thick) was purchased from KeJing MTI Corporation. The Cu foil was punched into 19 mm diameter circular disks before use. The Cu disks for CV measurements were soaked in 1 M HCl/H₂O (deoxygenation before use), then washed three times with deoxygenated water and ethanol, and then immediately transferred to the glovebox. The electrolytes with salt-to-solvent (LiFSI:DME) molar ratios of 1:10, 1:4, 1:2, 1:1.4, 1:1 and 1:0.7 (saturate concentration, 25 °C) were used as the working electrolytes.

### Electrochemical measurements

All electrochemical tests were performed using 2032-type coin cells at 25 °C. Cell fabrication was carried out in Ar-filled glovebox (<0.1 ppm O₂ and <0.1 ppm H₂O). Each cell used one piece of PE separator (16 μm in thickness, 19 mm in diameter) and 40 μL of electrolyte. Two pieces of Li foils of 450 μm thickness were used in Li/Li symmetric cells. In Li/Cu half cells, one piece of Cu disk was used as the working electrode,

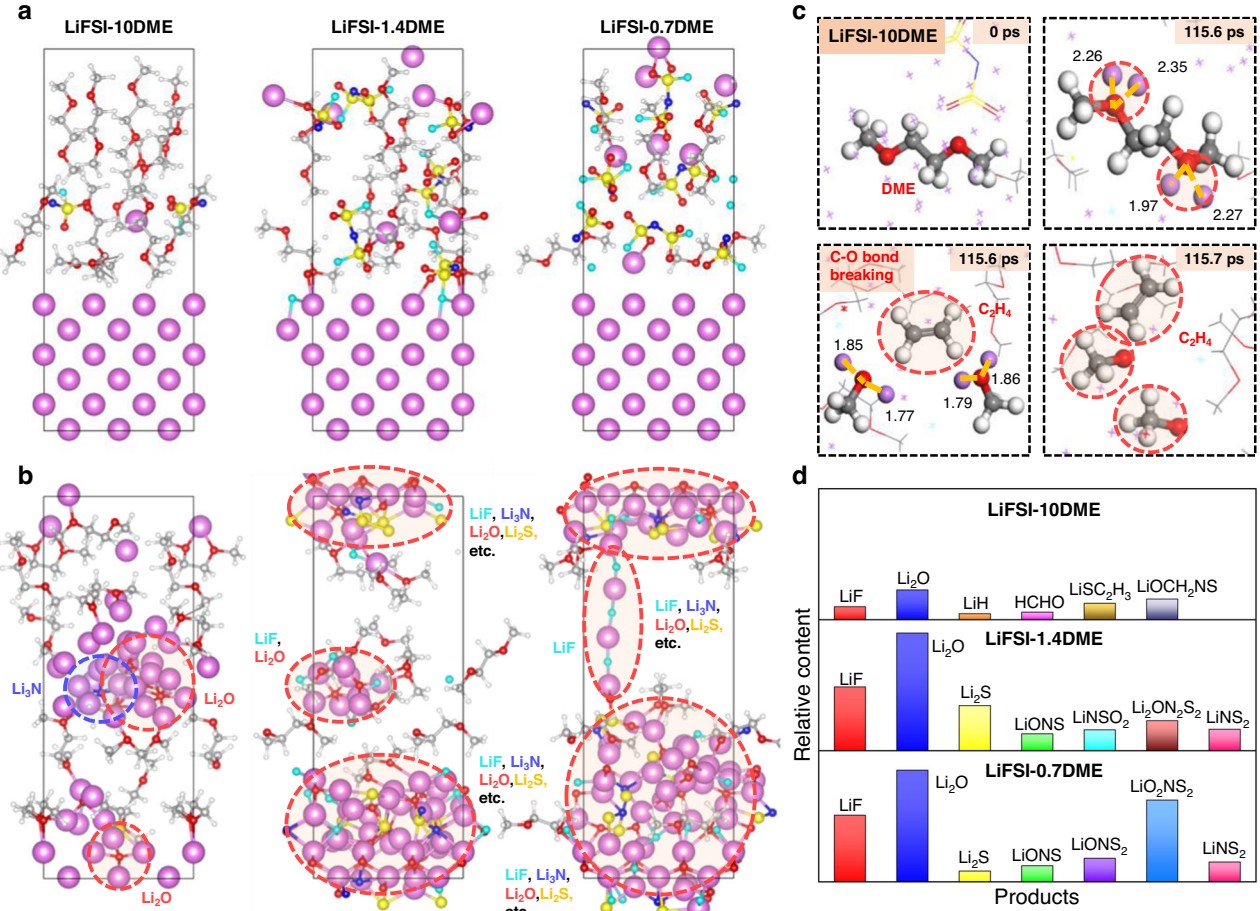

**Fig. 6 | Interfacial reaction and SEI formation on Li metal from HAIR-MD simulations.** The snapshots of **a** initial configurations and **b** interface reaction after 2.8 ns HAIR-MD simulations of LiFSI-xDME electrolytes (x = 10, 1.4, 0.7, respectively). **c** The reactive snapshots of the decomposition of DME solvent obtained from HAIR-MD simulations of the LiFSI-10DME electrolyte between 0.0-115.7 ps. **d** Relative contents of the main products after 2.8 ns HAIR-MD simulations. Color code in (**a**–**c**): H, white; Li, purple; C, gray; N, blue; O, red; F, cyan; S, yellow.

and one piece of Li foil (450 μm in thickness) was used as the counter electrode and the reference electrode. Li/Li and Li/Cu cells were tested on a Land or Neware battery test system at 25 °C. In the Li/Cu cells with Cu grid electrodes, the Cu grid was located between the PE separator and the Cu foil and was placed in the center of the electrode. For the long-term cycling of Li/Cu cells, 1 mAh cm⁻² of Li was plated on the Cu electrode and then stripped to 1 V at the current density of 0.5 mA cm⁻². Li/Li symmetric cells were plated and stripped at 0.5 mA cm⁻² for 1 mAh cm⁻². Cyclic voltammetry (CV) measurements were conducted in Li/Cu half cells at 1 mV s⁻¹ scan rate over the voltage window of 2.7–0 V vs. Li/Li⁺.

**Characterization**

Li deposition morphologies on Cu electrodes were imaged by scanning electron microscopy (SEM, FEI Apero, 2 kV). Fixed capacities (0.02–1 mAh cm⁻²) of Li were deposited on the Cu electrodes at the current density of 0.5 mA cm⁻² in Li/Cu cells. The cells were disassembled in the glovebox, and the Cu electrodes were rinsed with DME solvent. The test samples were transferred from the glovebox in an airtight container and quickly loaded into the SEM vacuum chamber.

The surface morphology of the electrodeposited Li and modulus distribution of the SEI layer were performed by atomic force microscopy (AFM, BRUKER, Dimension Icon, installed in Ar-filled glovebox) in peak-force quantitative nanomechanical mapping (QNM) mode with a sharp AFM tip (Bruker RTESPA-300). The mechanical parameters of

the probe were calibrated using polyethylene film standard samples, and the indentation depth was fixed at 2 nm. 1 mAh cm⁻² of deposited Li (0.5 mA cm⁻²) on the Cu electrode was used to prepare the AFM sample. The cells were disassembled in the glovebox with AFM, and the electrodes were rinsed with DME.

The chemical components of SEI on the plated Li were detected by depth-profiling X-ray photoelectron spectroscopy (XPS, ESCALAB 250Xi). To prepare the XPS sample, the Li was plated on Cu foil at 0.5 mA cm⁻² for 1 mAh cm⁻². The samples were rinsed with DME in the glovebox, and then sealed into the XPS dedicated transfer module before XPS measurement, without air contamination. Depth profiling was conducted by Ar ion sputtering (2 kV, 2 × 2 mm) on samples for 2 min.

The solvation structure of electrolytes and Li₂C₂ components on SEI was investigated by Renishaw inVia Raman Microscope with a spectral resolution of 1 cm⁻¹. The electrolytes (100 μL) were injected into a flat quartz airtight container with high light transmittance in the glovebox. The wavelength of the incident laser is 785 nm, and its focusing point position is set at about 500 μm below the inner wall of the quartz glass. The plated Li samples (0.5 mA cm⁻² for 1 mAh cm⁻²) were also placed in a high-transmittance flat quartz container for Raman mapping (50 × 50 μm, 21 × 21 pixels).

The Cu TEM grids for Cryo-EM were taken out from Li/Cu coin cells in a glovebox filled with argon, and then washed briefly with DME to remove electrolyte impurities. Cryo-transfer box was used to isolate water and oxygen during the whole process of sample transfer. Titan

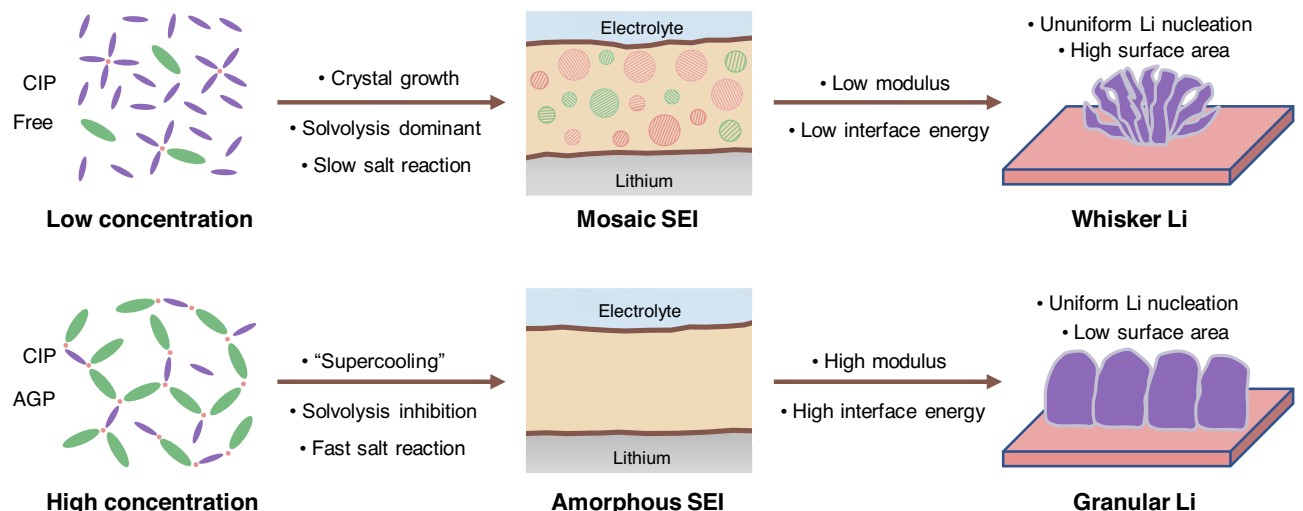

**Fig. 7 | Schematic illustration of Li morphology shaping.** Schematic of the relationships between the solvated electrolyte structure, SEI structure, and Li metal deposition morphology through the modulation of the salt concentration. CIP the contact-ion pair, AGP the aggregate ion pair.

Krios G3i Cryo-TEM (300 kV) equipped with a Falcon 3 Direct Electron Detector Camera was used to acquire TEM, STEM and EELS images of SEI at a low electron dose flux. The obtained Cryo-EM data were analyzed using Digital Micrograph software and materials database (Materials Project, Crystallography Open Database and ICDD PDF cards).

## Theoretical calculations

The models for HAIR-MD simulations consist of Li metal anode and electrolytes with different molar ratios. A 6-layer (3 × 3) supercell slab with the most stable Li(100) represents a Li metal anode. To achieve the desired concentrations and densities, 1 LiFSI and 10 DME molecules (molar ratio 1:10), 5 LiFSI and 7 DME molecules (molar ratio 1:1.4), 6 LiFSI and 4 DME molecules (molar ratio 1:0.7), are placed in a periodic box to represent the different concentrations. The final simulation periodic cell is approximately 10.5 × 10.5 × 26.5 Å.

The ab initio molecular dynamics (AIMD) and reactive molecular dynamics (RMD) are coupled with a homemade python shell to apply the HAIR simulations. The workflow of HAIR simulation consists of DFT optimization and AIMD equilibrium for the initial relaxation process to obtain reasonable initial configurations. After that, 0.5 ps AIMD followed by 5.0 ps RMD are conducted for one cycle HAIR simulation with a 10-time acceleration factor. After 50 cycle HAIR simulation (275 ps), each HAIR cycle includes 0.5 ps AIMD and 50.0 ps RMD because the mass transfer process is dominant, and the acceleration factor corresponds to 100-time with another 50 cycles (2525 ps).

The AIMD calculations are performed using Vienna Ab initio Simulation Package (VASP) at a version of 5.4.4 with the projector augmented wave (PAW) method and a plane wave basis set. A generalized gradient approximation (GGA) of the Perdew-Burke-Ernzerhof (PBE) functional is used in this work. As for the dispersion correction, the DFT-D3 method with Becke-Jonson damping is included in the calculations. The energy cut-off is set to 400 eV, and the gamma point of the Brillouin zone is considered to facilitate affordable computational cost. In addition, spin polarization is not included as it has an appreciable effect on the overall energies. The partial occupancies for each orbital are set with the first order Methfessel-Paxton scheme in the smearing width of 0.2 eV. The self-consistent electronic step is considered converged when the total energy change and eigenvalues change between two steps are smaller than $1e^{-4}$ eV. During the AIMD simulations, a 1 fs time-step is used with the hydrogen mass set to 2 atomic mass units in the canonical (NVT) ensembles at 300 K. For equilibrations, the velocities are scaled to the target temperature every 20 steps, and the canonical (NVT) ensemble was relaxed by using Nosé–Hoover thermostat with a damping parameter of 200 fs.

The RMD simulation was carried out using the USER-REAXC package in LAMMPS (12 Dec 2018) with a 0.25 fs time-step. The velocities are scaled to the target temperature (300 K) every 20 steps to guarantee good energy conservation while ensuring efficient convergence for collisions and smooth reactions. And the canonical (NVT) ensemble was relaxed using The Nosé–Hoover thermostat with a damping parameter of 50 fs. The ReaxFF force field parameters used in our RMD simulations are developed by Liu et al[47]., which is designed for the LiFSI system. The relative content of each product was evaluated by its relative mass content, which was normalized according to the total relative masses of 1920, 1943 and 2047 for LiFSI-10DME, LiFSI-1.4DME and LiFSI-0.7DME, respectively.

## Reporting summary

Further information on research design is available in the Nature Portfolio Reporting Summary linked to this article.

## Data availability

The data that support the findings of this study are available from the corresponding authors upon reasonable request.

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

## Acknowledgements

This study was supported by the National Key Research and Develop-ment Program of China (Grant No. 2022YFA1504102), the Strategic Priority Research Program of the Chinese Academy of Sciences (Grant No. XDB0450302), the National Natural Science Foundation of China (Grant Nos. 52225105, 22279127, 52072358, U21A2082). M.G. acknowledges supports by the Guangdong Fundamental Research Association (Grant No. 2022B1515120001), the National Natural Science Foundation of China (Grant No. 52273225), the Guangdong scientific program (Grant No. 2019QN01L057), the Shenzhen Science and Technology Program (Grant No. KQTD20190929173815000), the Guangdong Innovative and Entrepreneurial Research Team Pro-gram (Grant No. 2019ZT08C044). T.C. acknowledges supports by the National Natural Science Foundation of China (22173066), the National Key Research and Development Program of China (Grant No. 2022YFB2502200). This work was also supported by NIO Research Program (NRP).

## Author contributions

R.C. and S.J. conceived the idea and designed the project. Y.C. performed the measurements and analyzed the data. M.L. conducted the Cryo-EM experiments and data analysis under the supervision of M.G. Y.L. performed HAIR simulations and data analysis under the supervision of T.C. Y.C., W.L., and X.L. conducted SEM observation. Y.C., F.H., and Z.H. performed AFM experiments. Yu.C., X.M., X.R., and all authors discussed the results and commented on the paper. Y.C. wrote the manuscript, R.C., S.J., T.C., M.G., M.L., Y.L., and Y.J. revised the paper. Y.C., M.L., and Y.L. contributed equally to this work.

## Competing interests

The authors declare no competing interests.
