## [Peer Review File · Nature Communications]

Origin of dendrite-free lithium deposition in concentrated electrolytesREVIEWER COMMENTS

Reviewer #1 (Remarks to the Author):

The authors study the change in the SEIs chemical composition and morphology, as well as the lithium deposition morphology, as a function of LiFSI concentration in DME based electrolyte. They find that at low salt concentrations dendrites tend to form and that the SEI is made up of organic species and some crystalline nanoparticles. At high salt concentrations the SEI is richer in inorganic species and is largely amorphous as well as a more even Li deposition. The improved cyclability seen at high salt concentrations they attribute to the change in SEI structure and Li deposition.

This is a well carried out study, however, I question what findings are actually new. It is well known from the study of highly concentrated electrolytes that a high salt concentration leads to an SEI rich in inorganic species, and a low salt concentration to an SEI rich in organic species. It seems to me that the result with most novelty is then only the part about the SEI morphology (organic with nanoparticles at low concentration, and amorphous at high concentrations.) and their explanation for how this morphology comes about. But, what causes this is also were I think the evidence is the most lacking.

Major:

1. What is unique with this study? I think it must be more clearly highlighted what the novelty of the results are.

2. It seems the most novel results are with regards to the morphology and the super cooling, i.e. it is stated several times that the decomposition of anions is so fast that the decomposition products don't have time to form a true crystalline shape, and instead you get an amorphous SEI. Yet, I'm having a hard time identifying the evidence that the reduction of the anions is much faster than the reduction of the solvents, what is the evidence for this? It seems the current density used when characterizing the samples was always the same. As the electrolyte is much more viscous at high salt concentrations, and the overpotentials much higher, this might also have an effect. It seems to me that if a lower current density was used this amorphization might not happen and instead the system would form more crystalline particles. What is the authors opinion on this? To me it seems the simulations are blind to the applied current density. Was any investigation made into how the SEI morphology was affected by current density?

3. I have a few remarks regarding figure 2 in the supplementary information. First of all, I don't get the x-axis on figure 2b, now it reads as 1:10, 1:1.4, 1:0.7, 1:10, 1:1.4, 1:0.7. Should it not be 1:10, 1:4, 1:2, 1:1.4, 1:1, and 1:0.7 to fit with figure 2a? Second, it seems like the profiles were convoluted into only 2 parts for the spectra 1:10 1:4, 1:1 and 1:0.7, and into 3 parts for 1:2 and 1.4. I trust the results here are quite accurate, but deconvoluting this way sets up the result of no free solvents at very high concentrations, and no aggregates at low concentrations. I think it would be much more scientifically sound to always deconvolute into free, CIP and AGP, even if the AGP fraction is close to zero for low concentrations, and the free close to zero at high concentrations. But deconvoluting for free, CIP and AGP doesn't pre-assume this result.

Minor:

1. Plot of CE should be done in a narrow range as possible. Would also be nice if initial CE was written out in the figure.

2. Some language inconsistencies. For instance, I see Supper cooling, super cooling and supercooling. Sometimes the authors write Cu, sometimes copper or cooper, etc.

3. Choice of colour in some figures. For instance, the scale bars in figure 7 in supplementary are barely visible. Similarly, in figure 2h and 2i the dark blue on black background is almost impossible to see, maybe change the dark blue to white?

Reviewer #2 (Remarks to the Author):

In this manuscript, authors conducted in-depth analysis on the formation and development of SEI layer on Li metal surface using a prototypical LiFSI/DME electrolyte with different salt concentrations. Authors successfully observed the difference in the SEI composition and structure by varying the concentration of the electrolyte, which was supported by theoretical calculations. The difference in the SEI component was also correlated with the surface modulus mapping conducted by the AFM measurements to explain the improved cycle performance and prevention of dendritic lithium using highly concentrated solvent-in-salt-type electrolyte. The results are interesting and important for providing fundamental mechanisms of SEI formation/evolution and contributing to the development of the technology. However, there are several points that require clarification, and more experiments/analyses required to support the authors' conclusion before it can be considered for publication in Nature Communications.

1. Li-CE volcano trend was mentioned in the text, and it is clear by comparing the values for different concentrations. But it would be helpful for readers to notice the trend if there is a graph showing CE vs. concentration.
2. Authors mentioned that the presence of nanoparticles in SEI may reduce the strength and flexibility of the SEI. Evidence of fracture between the crystalline particle and the amorphous phase for low concentration electrolyte should be presented to support this. Additionally, Cryo-TEM after multiple cycles may be conducted to provide better picture of the SEI evolution and how different salt concentrations of the electrolyte contribute to the cycle stability.
3. HRTEM of the SEI on Li surface after Li plating was presented. However, detailed analysis on the change in the SEI morphology after stripping is missing to support the conclusion of this manuscript.
4. EELS data indicates the higher LiF content in the SEI formed in the LiFSI-1.4DME electrolyte. In what form does LiF exist in the SEI? If crystalline, why there is no crystalline particles observed by cryo-TEM?
5. Resolution of STEM-EELS elemental mapping maybe too low to support the authors' claim that elements are distributed uniformly across the SEI for both low and high concentration electrolytes, which is inconsistent with the HRTEM images of the SEI from low-concentration electrolyte having small crystals embedded in an amorphous matrix.
6. Simulation models used for HAIR MD simulations seem to be too small for analyzing the interfacial reactions. Moreover, all atoms in Li substrate were free to move causing it to dissolve into the electrolyte, which may not be representing the true interfacial reaction between Li surface and the electrolyte.
7. Fig. 5d shows relative contents of the main products, how are the data normalized with respect to which product in what concentration? Or was it normalized with the same product for each concentration? Please provide more details.
8. Higher modulus of the SEI formed on the highly concentrated electrolyte was attributed to the more inorganic components produced by the anionic decomposition. However, HAIR MD showed the formation of gases in lower concentration, which can lead to porous SEI. Which is the true cause of the increased modulus for higher concentrated electrolyte measured by the AFM experiments?
9. How does indentation depth affect the measured modulus of the SEI? Can it be correlated with the XPS depth-profile?

Response to the Reviewers Comments on the Manuscript NCOMMS- 22-41990

We highly appreciate the reviewers' constructive comments and insightful suggestions on this work. We have provided detailed answers and explanations to address the reviewers' questions. We have also incorporated most of the reviewers' comments and suggestions into the revised manuscript. The changes to the manuscript are marked **yellow** in this response and in the revised manuscript. Many thanks!

Reviewer #1 (Remarks to the Author):

The authors study the change in the SEIs chemical composition and morphology, as well as the lithium deposition morphology, as a function of LiFSI concentration in DME based electrolyte. They find that at low salt concentrations dendrites tend to form and that the SEI is made up of organic species and some crystalline nanoparticles. At high salt concentrations the SEI is richer in inorganic species and is largely amorphous as well as a more even Li deposition. The improved cyclability seen at high salt concentrations they attribute to the change in SEI structure and Li deposition.

This is a well carried out study, however, I question what findings are actually new. It is well known from the study of highly concentrated electrolytes that a high salt concentration leads to an SEI rich in inorganic species, and a low salt concentration to an SEI rich in organic species. It seems to me that the result with most novelty is then only the part about the SEI morphology (organic with nanoparticles at low concentration, and amorphous at high concentrations.) and their explanation for how this morphology comes about. But, what causes this is also where I think the evidence is the most lacking.

Response: We appreciate the reviewer for the constructive comments. Your professional suggestions and comments are very helpful for us to improve the quality of our manuscript. We thank the reviewer for the comment "this is a well carried out

study”. We also understand the reviewer’s concerns about the novelty and completeness of our work, and we have addressed these concerns by further re-emphasizing the highlights of our findings. In addition, we have also conducted additional experiments and analysis as the reviewer suggested and revised the manuscript accordingly.

Major 1: What is unique with this study? I think it must be more clearly highlighted what the novelty of the results are.

Response: We thank the reviewer for the critical comments and suggestions, and we understand the reviewer’s concerns about the novelty of our work. High concentration electrolyte (HCE) as one of the most effective electrolyte engineering strategies to stabilize lithium batteries has been gradually established since 1985 (J. Electrochem. Soc., **1985**, *132*, 364; J. Electrochem. Soc., **2015**, *162*, A2406–A2423). Jeong et al. first applied concentrated electrolytes to Li metal anodes to achieve higher Li Coulombic efficiencies and less dendritic deposition (Electrochem. Commun., **2008**, *10*, 635–638). Qian et al. used 4M LiFSI/DME electrolyte to enable high-rate cycling and high Coulombic efficiency of Li metal anode without dendrite growth, which has inspired extensive research on highly concentrated electrolytes on lithium metal batteries. However, the microstructures of SEIs and their formation mechanism in electrolytes with different concentrations lack systematic research. In addition, the origin of dendrite/dendrite-free formation in low/high concentration electrolytes still remains elusive and unclear. A unified descriptor is still lacking for the effects of SEIs with different structures and compositions on Li deposition behavior, although many debates persist including homogeneity (Nature, **2019**, *572*, 511-515), mechanical stability (Nat. Energy, **2019**, *4*, 664-670), ionic conductance (Nat. Commun., **2015**, *6*, 6362), SEI swelling (Science, **2022**, *375*, 66-70), fluorinated interphases (Chem, **2018**, *4*, 174-185), inorganic-rich interphases (Nat. Mater., **2022**, *21*, 445-454), etc. These issues are crucial but not well established for the study of lithium metal anodes, which is also the focus of our work.

The comparison of different SEI structures in the literature often comes from different electrolyte systems or the introduction of new additives, so it is not simply a change in the solvation structure, but also the difference in the interface chemical compositions. Therefore, we selected a model electrolyte composing LiFSI/DME combination to systematically study the origin of dendrite/dendrite-free formation. We also **provide clear pictures and understandings of the SEI structures in low-concentration and high-concentration electrolytes, which have not been established in previous studies.** Our work attempts to **correlate the solvation structure of the electrolyte to the SEI formation process and the regulation mechanism of the SEI microstructure on Li metal deposition.** For the SEI formation, we believe that the decomposition speed of anions is very critical to the formation of SEI structure. Under high concentration conditions, the anion-dominated solvation structure accelerates the decomposition process of anions, and such rapid “supercooling” state leads to the formation of amorphous SEI. Whereas a slower salt decomposition process and non-negligible solvolysis in the diluted electrolytes can facilitate the inoculating crystals of the inorganic components, thus yielding organic-rich SEI (with more deleterious Li_2C_2 , ACS Energy Lett., **2022**, 7, 3458–3466) with a mosaic structure. For the Li metal deposition regulation, we found that the high-interface-energy amorphous SEI formed in concentrated electrolytes endows the dendrite-free Li deposition. In contrast, the organic-rich mosaic SEI formed in dilute electrolytes poses a lower specific interface energy, resulting in the formation of whisker Li or dendrite Li with high surface area. Our fundamental understandings will be helpful to tackle the long-standing question on the origin of lithium dendrite formation and guide the rational design of high-performance electrolytes for advanced lithium metal batteries.

We have provided additional experiments and analyzes to improve the novelty and completeness of our work. The reactivities of the salt and solvent can be distinguished respectively by electrochemical CV test (Figure R2). A new figure (Figure R1) as Figure 2 in the main text of the revised manuscript provided to demonstrate the generality of this concentration-induced SEI structural difference. In addition, to

strengthen the novelty, we have re-edited the following text in the Introduction as follows:

“However, no consistent conclusions have emerged to explain the intrinsic motivation for forming these SEI structures. The comparison of different SEI structures in previous studies has been carried out in different electrolyte systems or new additives^{24,35,39,40}, which results in not only changes in solvation structures but also differences in interfacial chemical compositions.

...Our fundamental studies on the origin of dendrite/dendrite-free formation in low/high concentration electrolytes can deepen the understanding of electrolyte solvation structure design, SEI growth behavior, and their modulation on Li deposition morphology.”

Figure R1. SEI structures formed at a low current density or after multiple cycles. Cryo-TEM images of SEI formed in (a, c) LiFSI-10DME and (b, d) LiFSI-1.4DME electrolytes (a, b) at a low current density of 0.1 mA cm^{-2} and (c, d) on the freshly

deposited Li metal after 5 cycles. The right sides of a and c show HRTEM images and corresponding fast Fourier transform (FFT) of selected regions.

Major 2: It seems the most novel results are with regards to the morphology and the super cooling, i.e. it is stated several times that the decomposition of anions is so fast that the decomposition products don't have time to form a true crystalline shape, and instead you get an amorphous SEI. Yet, I'm having a hard time identifying the evidence that the reduction of the anions is much faster than the reduction of the solvents, what is the evidence for this? It seems the current density used when characterizing the samples was always the same. As the electrolyte is much more viscous at high salt concentrations, and the overpotentials much higher, this might also have an effect. It seems to me that if a lower current density was used this amorphization might not happen and instead the system would form more crystalline particles. What is the authors opinion on this? To me it seems the simulations are blind to the applied current density. Was any investigation made into how the SEI morphology was affected by current density?

Response: We thank the reviewer for the good questions and constructive comments. In response to the reviewer's comments, we provided electrochemical cyclic voltammetry (CV) tests to directly demonstrate the difference in anion reactivity in electrolytes with different concentrations. In addition, we performed Cryo-EM measurements to identify the microstructure of the SEI formed at a low current density.

During the galvanostatic deposition of Li metal on Cu, the decomposition of salt and solvent at the interface occurs almost simultaneously, which is difficult to decouple. However, if the potential is controlled, the reactivities of the salt and solvent can be distinguished respectively by the CV test before Li metal deposition. As shown in Figure R2, the reduction peaks at ~ 1.7 V can be attributed to the reductive decomposition process of FSI⁻ anion (J. Mater. Chem. A, **2020**, 8, 3459–3467), and the reductive decomposition process of the solvent is shown as a subsequent sloping curve. In the lower concentration electrolytes (LiFSI-10DME and LiFSI-4DME), the ~ 1.7 V

reduction peak is absent and indistinguishable. In the LiFSI-2DME, the CV reduction curve has a slight fluctuation at ~ 1.7 V. Whereas under high concentration conditions (LiFSI-xDME, $x \geq 1.4$), the peak intensity of the anion reduction peak gradually increases with the increase in concentration (Figures R2 and R3). In addition, the current densities for solvolysis at high concentrations is lower than that at low concentrations. These results directly indicate that anions are more easily decomposed with increasing concentration and the solvolysis is partially suppressed, in line with the cryo-TEM, XPS, Raman mapping and theoretical simulation results.

As the reviewers pointed out, the overpotentials for Li metal plating and stripping gradually increases with the increasing concentration (Figure R4). The difference in these overpotentials mainly comes from the increase in impedance caused by the high viscosity of high concentration electrolytes. The increase of the overpotential may have a certain influence on the electrochemical reaction process at the lithium metal interface. Therefore, following the reviewer's suggestion, we characterized the Li metal deposition morphology and SEI microstructure under lower current density conditions (0.1 mA cm^{-2}) by cryo-electron microscopy (cryo-EM). We choose the representative electrolytes of LiFSI-10DME and LiFSI-1.4DME for our study, and the Li deposition capacity is still fixed at 0.5 mAh cm^{-2} .

As shown in Figure R5 a-c, the low magnification cryo-TEM images of deposited Li with the LiFSI-10DME electrolyte show that Li has a more complete and dense morphology at a lower current density. Most importantly, the SEI formed on the surface of Li metal still maintains the mosaic structure (Figure R5 d-f), which indicates that the inorganic components can still grow nanocrystals in the low-concentration electrolyte even under low current density conditions. In the high-concentration LiFSI-1.4DME electrolyte, the deposited Li displays a dendrite-free and granular Li morphology (Figure R6 a-c). Interestingly, at the low current density of 0.1 mA cm^{-2} , the SEI under high-concentration conditions still exhibits amorphous structure (Figure R6 d-f). This indicates that the amorphization process at the interface still occurs at low current densities in high-concentration electrolytes without producing inorganic nanocrystals.

In addition, the average thickness of the SEI in LiFSI-1.4DME (8.0 nm) is significantly lower than that of LiFSI-10DME (12.2 nm), which further demonstrates the effective passivation of the interface in highly concentrated electrolytes.

We have included the electrochemical CV data and the cryo-TEM morphological characterization at a lower current density into the revised manuscript as Supplementary Figures 20 and 21, Figure 2 a, b and Supplementary Figures 11 and 12, respectively, and the related description in the revised manuscript and is copied as follows for easy review.

“The electrochemical reactivities of the FSI⁻ anion and DME solvent can be distinguished by the cyclic voltammetry (CV) tests at a scan rate of 1 mV s⁻¹ on Cu electrodes before Li metal deposition (2.7-0 V vs. Li/Li⁺). As shown in Supplementary Fig. 20, the reduction peaks at ~1.7 V can be attributed to the reductive decomposition process of FSI⁻ anion⁴⁵, and the reductive decomposition process of the DME solvent is shown as a subsequent sloping curve. In the lower concentration electrolytes (LiFSI-10DME and LiFSI-4DME), the ~1.7 V reduction peak is absent and indistinguishable. In the LiFSI-2DME electrolyte, the CV reduction curve has a slight fluctuation at ~1.7 V. Whereas under high concentration conditions (LiFSI-xDME, x≥1.4), the peak intensity of the anion reduction peak gradually increases with the increase in concentration (Supplementary Figs. 20 and 21). In addition, the current densities for solvolysis at high concentrations is lower than that at low concentrations. These results directly indicate that anions are more easily decomposed with increasing concentration and the solvolysis is partially suppressed, in line with the cryo-TEM, XPS, Raman mapping and theoretical simulation results.”

“We also characterized the Li metal deposition morphology and SEI microstructure under lower current density conditions (0.1 mA cm⁻²) by cryo-TEM. We choose the representative electrolytes of LiFSI-10DME and LiFSI-1.4DME for our study, and the Li deposition capacity is still fixed at 0.5 mAh cm⁻². As shown in Supplementary Fig. 11 a-c, the low-magnification cryo-TEM images of deposited Li with the LiFSI-

10DME electrolyte display a denser morphology at a lower current density. We also note that the SEI still maintains the mosaic structure (Figure 2a, Supplementary Fig. 11 d-f). In the high-concentration LiFSI-1.4DME electrolyte, the deposited Li displays a dendrite-free and granular Li morphology (Supplementary Fig. 12 a-c). Interestingly, at the low current density of 0.1 mA cm^{-2} , the SEI still exhibits an amorphous structure (Fig. 2b, Supplementary Fig. 12 d-f). In addition, the average thickness of the SEI in LiFSI-1.4DME (8.0 nm) is significantly lower than that of LiFSI-10DME (12.2 nm), which further demonstrates the effective passivation of the interface in highly concentrated electrolytes.”

Figure R2. First cycle CV curves of Li/Cu half cells with different electrolytes over

the voltage range from 2.7 V to 0 V.

Figure R3. Relative peak areas of the reduction peaks of the CV curves at ~1.7 V.

Figure R4. Overpotential for plating and stripping of Li in Li||Cu cells using various salt-to-solvent molar ratios of electrolytes.

Figure R5. Cryo-TEM images of 0.5 mAh cm^{-2} deposited Li (a-c) and SEI (d-f) in the LiFSI-10DME electrolyte. The current density is 0.1 mA cm^{-2} .

Figure R6. Cryo-TEM images of 0.5 mAh cm^{-2} deposited Li (a-c) and SEI (d-f) in the LiFSI-1.4DME electrolyte. The current density is 0.1 mA cm^{-2} .

Major 3: I have a few remarks regarding figure 2 in the supplementary information. First of all, I don't get the x-axis on figure 2b, now it reads as 1:10, 1:1.4, 1:0.7, 1:10, 1:1.4, 1:0.7. Should it not be 1:10, 1:4, 1:2, 1:1.4, 1:1, and 1:0.7 to fit with figure 2a? Second, it seems like the profiles were convoluted into only 2 parts for the spectra 1:10 1:4, 1:1 and 1:0.7, and into 3 parts for 1:2 and 1.4. I trust the results here are quite accurate, but deconvoluting this way sets up the result of no free solvents at very high concentrations, and no aggregates at low concentrations. I think it would be much more scientifically sound to always deconvolute into free, CIP and AGP, even if the AGP fraction is close to zero for low concentrations, and the free close to zero at high concentrations. But deconvoluting for free, CIP and AGP doesn't pre-assume this result.

Response: We thank the reviewer for the careful review and constructive suggestions. We have corrected the mistakes in Supplementary Figure 2. According to the reviewer's suggestions, we refitted all Raman spectra according to Free, CIP and AGP. The fitting results and information are shown in Figure R7 and Table R1. The refitted results are indeed more plausible and we have corrected it in the revised manuscript.

Figure R7. Raman spectra of LiFSI/DME electrolytes with various molar ratios. (b)

Fitting results of free FSI-, CIP and AGP ratios in LiFSI/DME electrolytes.

Table R1. Detailed information of the Raman fitted peaks.

	Curve	Centre	Width	Height	Area	Percentage
10	Free	717.5	13.6271	53190	860249	67.893
	CIP	728.2	14.6308	18722.6	291587	23.013
	AGP	742.201	16.0958	6725.56	115232	9.094
4	Free	717.6	13.7871	99365	1591520	53.228
	CIP	728.5	16.8986	56070.3	1008590	33.732
	AGP	744.771	15.8441	17805	389914	13.040
2	Free	717.65	15.9885	20975.2	356981	6.803
	CIP	729.5	20.3624	116181	2518240	47.992
	AGP	746	22.1594	80207.9	2372040	45.205
1.4	Free	717.7	1.05459	1000	1123	0.016
	CIP	730	18.0819	43820.8	843443	12.379
	AGP	746.3	26.4171	185836	5968880	87.604
1	Free	720	0.44986	1000	479	0.002
	CIP	734	13.0319	28654.1	397491	2.049
	AGP	750.884	26.4313	594248	19001900	97.949
0.7	Free	720	0.449753	1000	479	0.003
	CIP	736	0.527173	103355	72705	0.425
	AGP	754.242	25.1427	547165	17039100	99.572

Minor 1: Plot of CE should be done in a narrow range as possible. Would also be nice if initial CE was written out in the figure.

Response: We thank the reviewer for the careful review and good suggestions, and we have corrected and added in the revised manuscript, and the corrected figure is shown in Figure R8.

Figure R8. Long-term cycling Coulombic efficiencies of Li/Cu half cells in LiFSI-xDME electrolytes ($x=10, 4, 2, 1.4, 1, 0.7$, respectively). The average CEs are calculated from the first 500 cycles.

Minor 2: Some language inconsistencies. For instance, I see Supper cooling, super cooling and supercooling. Sometimes the authors write Cu, sometimes copper or cooper, etc.

Response: We thank the reviewer for the careful review, and we have made all the revisions in the revised manuscript.

Minor 3: Choice of colour in some figures. For instance, the scale bars in figure 7 in supplementary are barely visible. Similarly, in figure 2h and 2i the dark blue on black background is almost impossible to see, maybe change the dark blue to white?

Response: We thank the reviewer for the careful review and the good advice. We have corrected these figures in the revised manuscript, as shown in Figures R9-R11.

Figure R9. SEM images of the Cu electrodes after one plating/stripping cycle (0.5 mA cm^{-2} for 1 mAh cm^{-2}) in different electrolytes.

Figure R10. Cryo-TEM images of deposited Li in LiFSI- x DME electrolytes ($x=10, 1.4, 0.7$, respectively). The current density is 0.5 mA cm^{-2} , and the capacity is 0.5 mAh cm^{-2} .

Figure R11. Cryo-STEM ADF images and EELS mapping images of SEI on Li metal in the (a) LiFSI-1.4DME and (b) LiFSI-0.7DME electrolytes.

Reviewer #2 (Remarks to the Author):

In this manuscript, authors conducted in-depth analysis on the formation and development of SEI layer on Li metal surface using a prototypical LiFSI/DME electrolyte with different salt concentrations. Authors successfully observed the difference in the SEI composition and structure by varying the concentration of the electrolyte, which was supported by theoretical calculations. The difference in the SEI component was also correlated with the surface modulus mapping conducted by the AFM measurements to explain the improved cycle performance and prevention of dendritic lithium using highly concentrated solvent-in-salt-type electrolyte. The results are interesting and important for providing fundamental mechanisms of SEI formation/evolution and contributing to the development of the technology. However, there are several points that require clarification, and more experiments/analyses required to support the authors' conclusion before it can be considered for publication in Nature Communications.

Response: We thank the reviewer for the constructive comments on this work. We have added more experiments and analyses to improve the quality of our manuscript according to the reviewer's suggestions. Our point-by-point reply to the comments in detailed as follows.

1. Li-CE volcano trend was mentioned in the text, and it is clear by comparing the values for different concentrations. But it would be helpful for readers to notice the trend if there is a graph showing CE vs. concentration.

Response: We thank the reviewer for the good suggestion. We have added the Li-CE volcano curve as Supplementary Figure 5 in the revised manuscript, as shown in Figure R12.

Figure R12. Average Li CEs of the first 500 cycles for various electrolytes.

2. Authors mentioned that the presence of nanoparticles in SEI may reduce the strength and flexibility of the SEI. Evidence of fracture between the crystalline particle and the amorphous phase for low concentration electrolyte should be presented to support this. Additionally, Cryo-TEM after multiple cycles may be conducted to provide better picture of the SEI evolution and how different salt concentrations of the electrolyte contribute to the cycle stability.

Response: We thank the reviewer for the insightful comments and constructive suggestions. Following the reviewer's suggestion, we attempted to directly observe the evidence of fracture between the crystalline grains and the amorphous phase by cryo-TEM. The observation of this phenomenon itself is very difficult, and we did not find it in the end. We therefore decided to delete the relevant description in the revised manuscript. However, in the process of observing the SEI after stripping, we did observe the phenomenon of SEI rupture under low concentration conditions (Figure R15e).

We used the cryo-TEM to characterize the Li deposition morphology and SEI structure after multiple cycles as suggested by the reviewer. The low-magnification cryo-TEM images show that the deposited Li in the LiFSI-10DME electrolyte after 5

cycles exhibits mossy Li and dendrite morphology (Figure R13 a-c) and a large number of deflated SEI husks (Figure R13 a, d-f). The existence of deflated SEI husks indicates that the subsequent Li metal is difficult to deposit into the SEI shells, so more electrolyte is consumed to form new SEI, resulting in the accumulation of SEI husks after multiple cycles, which in turn leads to a sharp increase in the thickness of the Li metal anode (Supplementary Figure 8m). More importantly, the SEI formed on the deposited Li metal still exhibits a mosaic structure after multiple cycles (Figure R13 g-i). In addition, the thickness distribution of SEI also shows obvious inhomogeneity, 9.5 nm, 21.1 nm, and 22.7 nm in Figure R13 g-i, respectively. This indicates that two or more layers of SEI may accumulate at the Li interface during cycling, further confirming the interphasial instability in low concentration electrolytes.

In sharp contrast, in the high concentration LiFSI-1.4DME electrolyte, the deposited Li after 5 cycles display granular and columnar morphology with few dendrites. In addition, the deflated SEI husks are absent (Figure R14 a-f), which indicates that Li metal can re-grow inside the SEI void shell. Surprisingly, even after multiple cycles, the amorphous structure SEI can still be kept, and a dense SEI with an average thickness of 8.4 nm is formed (Figure R14 g-i). These results further indicate that the amorphous SEI formed in high concentration electrolytes has better stability and Li metal compatibility than low concentration electrolytes.

We have included the cryo-TEM results after multiple Li cycles into the revised manuscript as Figure 2 c, d and Supplementary Figures 15 and 16, and the related description in the revised manuscript and is copied as follows for easy review.

“After the 5th cycle of Li plating and stripping, the newly deposited Li in the LiFSI-10DME electrolyte exhibits severe Li dendrite morphology (Supplementary Fig. 15 a-c) with a large number of deflated SEI husks (Supplementary Fig. 15 a, d-f). The existence of deflated SEI husks indicates that the subsequent Li metal is difficult to deposit into the SEI shells. More electrolyte is consumed to form the new SEI, resulting in the accumulation of SEI husks after multiple cycles, which in turn leads to a sharp increase in the thickness of the Li metal anode (Supplementary Fig. 9m). More

importantly, the SEI formed on the deposited Li metal still exhibits a mosaic structure after multiple cycles (Fig. 2c, Supplementary Fig. 15 g-i). In addition, the thickness distribution of SEI also shows obvious inhomogeneity (9.5 nm, 21.1 nm, 22.7 nm, Fig. 2c and Supplementary Fig. 15 g-i). This indicates that two or more layers of SEI may accumulate at the Li interface during cycling, further confirming the interphasial instability in low concentration electrolytes. In sharp contrast, in the LiFSI-1.4DME electrolyte, the deposited Li after 5 cycles display granular and columnar morphology with few dendrites. In addition, the deflated SEI husks are absent (Supplementary Fig. 16 a-f), which indicates that Li metal can re-grow inside the SEI void shells. Surprisingly, even after multiple cycles, the amorphous structure SEI can still be kept, and a dense SEI with an average thickness of 8.4 nm is formed (Fig. 2d, Supplementary Fig. 16 g-i). These results further indicate that the amorphous SEI formed in high concentration electrolytes endows better stability and Li metal compatibility than low concentration electrolytes.”

Figure R13. Cryo-TEM images of 0.5 mAh cm⁻² deposited Li (a-c), deflated SEI husks (d-f) and SEI (g-i) after 5 cycles in the LiFSI-10DME electrolyte.

Figure R14. Cryo-TEM images of 0.5 mAh cm⁻² deposited Li (a-f) and SEI (g-i) after 5 cycles in the LiFSI-1.4DME electrolyte.

3. HRTEM of the SEI on Li surface after Li plating was presented. However, detailed analysis on the change in the SEI morphology after stripping is missing to support the conclusion of this manuscript.

Response: Thank the reviewer for this good suggestion. We carried out Cryo-TEM experiments to characterize the Li stripping morphologies in the LiFSI-10DME and LiFSI-1.4DME electrolytes, as shown in Figures R15 and R16, respectively. Due to the electronic sensitivity of the SEI, the deflated empty SEI husks after Li stripping is difficult for HRTEM characterization. However, the low magnification cryo-TEM images can also provide important and useful information. When using the LiFSI-10DME electrolyte, the stripped Li metal exhibits obvious inhomogeneity with a large

amount of dead Li (Figure R15 a-d). The dendrites are partially stripped, and notched structures appear at the kinks of the dendrite (Figure R15 c and d), which is consistent with the results in the literature (Joule, **2018**, 2, 2167–2177). And the cracked SEI appears at the notched region as shown in the enlarged cryo-TEM image of Figure R15e. By contrast, under the high concentration condition, deflated empty SEI husks display a complete and unbroken morphology without visible dead Li on the Cu mesh (Figure R16), which corresponds to its high Li CE.

We have included the cryo-TEM results after Li stripping into the revised manuscript as Supplementary Figures 13 and 14, and the related description in the revised manuscript and is copied as follows for easy review.

“To evaluate the robustness of the SEI with different structures, we further carried out the cryo-TEM observation of the SEI morphology after stripping. When using the LiFSI-10DME electrolyte, the stripped Li metal exhibits obvious inhomogeneity with a large amount of dead Li (Supplementary Fig. 13 a-d). The dendrites are partially stripped, and notched structures appear at the kinks of the dendrites (Supplementary Fig. 13 c, d). As clearly shown in the enlarged cryo-TEM image of Supplementary Fig. 13e, the crack of SEI appears at the notched region. By contrast, under the high concentration condition, deflated empty SEI husks display a complete and unbroken morphology without visible dead Li (Supplementary Fig. 14), which corresponds to its high Li CE.”

Figure R15. Cryo-TEM images of the dead Li and SEI after stripping after 1 cycle in

the LiFSI-1.4DME electrolyte.

Figure R16. Cryo-TEM images of the SEI after stripping after 1 cycle in the LiFSI-1.4DME electrolyte.

4. EELS data indicates the higher LiF content in the SEI formed in the LiFSI-1.4DME electrolyte. In what form does LiF exist in the SEI? If crystalline, why there is no crystalline particles observed by cryo-TEM?

Response: We thank the reviewer for this good comment. LiF nanocrystals were not observed in our electrolyte system through the cryo-TEM experiments. Therefore, we think that LiF exists in an amorphous form in the SEI. In fact, the LiF component observed by cryo-TEM in literatures also mainly exhibits an amorphous structure, and the crystalline particles in SEI are mainly Li_2O and Li_2CO_3 (Science, **2017**, 358, 506–510; Joule, **2018**, 2, 2167–2177; Adv. Mater., **2021**, 33, 2100404; Adv. Mater. **2022**, 34, 2108252). Some literatures found the existence of a small amount of LiF crystals (Nature, **2019**, 572, 511–515; Nat. Energy, **2020**, 5, 534–542; Matter, **2021**, 4, 3741–3752). We believe that the amorphous component is the most important component in the SEI, and the role of the amorphous LiF component in the SEI needs

further study.

5. Resolution of STEM-EELS elemental mapping maybe too low to support the authors' claim that elements are distributed uniformly across the SEI for both low and high concentration electrolytes, which is inconsistent with the HRTEM images of the SEI from low-concentration electrolyte having small crystals embedded in an amorphous matrix.

Response: We thank the reviewer for the careful review and constructive comment. We provided STEM-EELS mapping images of SEI formed in the LiFSI-10DME electrolyte with the highest possible spatial resolution, as shown in Figure R17. Unfortunately, we did not find obvious elemental inhomogeneity. This is related to the thin thickness of SEI (~10 nm) in our cryo-TEM results, and the spatial resolution of EELS is not enough to distinguish crystal particles with a diameter of <5 nm. In addition, from the XPS and EELS spectra, the amorphous matrix also contains a large amount of Li_2CO_3 and Li_2O components, which indicates that the difference in element content between the amorphous matrix and the crystalline particles cannot be distinguished by the EELS mapping.

Figure R17. The cryo-STEM ADF image and EELS mapping images of SEI on Li metal in the LiFSI-10DME electrolyte.

6. Simulation models used for HAIR MD simulations seem to be too small for analyzing the interfacial reactions. Moreover, all atoms in Li substrate were free to move causing it to dissolve into the electrolyte, which may not be representing the true interfacial reaction between Li surface and the electrolyte.

Response: We greatly appreciate your valuable comment. The simulation models used

in this work are as same as in Balbuena's work (Ref1 and Ref2), which were modelled for the simulation of reductive reactions between electrolyte and Li metal anode, and two bottom of Li atoms are fixed in this work. In order to take the size effects into consideration, a larger model is also tested. We found that the computational cost is increased by 6 times as the model is doubled. In Balbuena's work, tens of picoseconds of AIMD are simulated to investigate the initial reduction of electrolytes. In our HAIR simulation, long-time simulations beyond nanosecond level are performed, which is benefited from the hybrid dynamics, and it takes one month to simulate the nanosecond level using a 32-cores node. Thus, a double-size model does not make much difference in simulating the chemical reaction process, and the exponential increase of computational cost exceeds the time cost we can afford.

To further conduct a large simulation model, 500 molecules DME with 50 LiFSI with $12 \times 12 \times 10$ Li(100) are placed in a $4.8 \times 4.8 \times 12.3$ nm box, and the geometry optimization and NVT reactive molecular simulations (RMD) are performed using LAMMPS, which lasts for 1 ns. As shown in Figure R18, the decomposition of FSI⁻ (mainly S-F and N-S bond breaking) are observed during MD simulation, and only several DME experiences reductive reactions, which is almost consistent with the results in our HAIR simulation.

Figure R18. The snapshot of the interface reactions after 1 ns HAIR-MD simulation of the LiFSI-10DME electrolyte in a large simulation model.

References

Ref1: Camacho-Forero, L. E.; Smith, T. W.; Bertolini, S.; Balbuena, P. B. Reactivity at the Lithium–Metal Anode Surface of Lithium–Sulfur Batteries. *J. Phys. Chem. C* 2015, 119, 26828–26839.

Ref2: Camacho-Forero, L. E.; Smith, T. W.; Balbuena, P. B. Effects of High and Low Salt Concentration in Electrolytes at Lithium–Metal Anode Surfaces. *J. Phys. Chem. C* 2017, 121, 182–194.

7. Fig. 5d shows relative contents of the main products, how are the data normalized with respect to which product in what concentration? Or was it normalized with the same product for each concentration? Please provide more details.

Response: We thank the reviewer for this good question. In fact, we do not normalize the final products, the final products of different systems are obtained from the final configuration of HAIR simulations, we used the total relative molecular mass of the products to evaluate the content of each product. To further explore the product distributions of different concentrations, we normalized the final products with the total relative molecular mass of each model (the total masses of these models are 1920, 1943 and 2047 for 1:10, 1: 1.4 and 1:0.7), as shown in Figure R19. We found that the product distributions show no significant difference after normalization because of the similar total mass of each model. The relative content of one product is defined as the ratio of the total relative molecular mass of the product obtained after simulation to the total relative molecular mass of the model.

We have corrected this figure in Figure 6d and provided more details in the Theoretical Calculations of Methods in the revised manuscript and is copied as follows for easy review.

“The relative content of each product was evaluated by its relative mass content, which was normalized according to the total relative masses of 1920, 1943 and 2047 for LiFSI-10DME, LiFSI-1.4DME and LiFSI-0.7DME, respectively.”

Figure R19. Relative contents of the main products after 2.8 ns HAIR-MD simulations.

8. Higher modulus of the SEI formed on the highly concentrated electrolyte was attributed to the more inorganic components produced by the anionic decomposition. However, HAIR MD showed the formation of gases in lower concentration, which can lead to porous SEI. Which is the true cause of the increased modulus for higher concentrated electrolyte measured by the AFM experiments?

Response: We thank the reviewer for the insightful comments. We believe that the most important factor affecting the modulus of SEI comes from the chemical composition of SEI itself, more specifically, the ratio of inorganic components to organic components in SEI. It is generally believed that the inorganic components (Li₂O, Li₂CO₃, LiF, Li₂S, Li₃N, etc.) in the SEI have a higher modulus to suppress Li dendrites, while the organic components with a low modulus (polymer, RLi, RO₂Li, etc.) make the SEI with a certain flexibility, but has poor mechanical properties and dissolution problems.

Our simulation results show that the decomposition of DME at low concentrations is not negligible, and the decomposition of the solvent may generate gases, but these

gases are more likely to accumulate instead of being embedded in the SEI to participate in the formation of the SEI structure. Porous cavities in the SEI were not observed in cryo-TEM images. The cryo-TEM results showed that the SEI formed in the low concentration electrolyte was still complete and dense, except for the special mosaic structure. In addition, the AFM morphology and homogeneous modulus distribution did not reveal evidence for the existence of such a porous structure SEI. So, we don't think the porosity of SEI is the main factor affecting the modulus in our research system.

And we have corrected the related expression in Page 18 in the revised manuscript and is copied as follows for easy review.

Second, the reduced gas productions, such as C_2H_4 , mitigates the structural damage of the SEI due to gas release.

9. How does indentation depth affect the measured modulus of the SEI? Can it be correlated with the XPS depth-profile?

Response: We thank the reviewer for the good questions. We performed QNM modulus analysis with different indentation depths as suggested by the reviewer. As shown in Figure R20, the SEI formed in the LiFSI-10DME electrolyte possesses a low average modulus of ~ 0.83 Gpa at an indentation depth of 2.1 nm. As the indentation depth increases, the average modulus of the formed SEI tends to increase gradually. In the high-concentration LiFSI-1.4DME electrolyte, the average modulus also increases with the indentation depth (Figure R21). Furthermore, at similar indentation depths, the SEI modulus in the LiFSI-1.4DME electrolyte is significantly higher than the low concentration counterpart (Figure R22).

These results are in line with the XPS depth-profile. The XPS analysis indicates that the organic components are mainly distributed in the outer layer of the SEI and inorganic components are in the inner layer of SEI. Therefore, the inorganic-rich SEI inner layer should have a higher modulus than the organic-rich outer layer, as confirmed by our AFM modulus analysis.

Figure R20. AFM height images (a-c) and modulus distribution images (d-f) with different indentation depths. 1 mAh cm^{-2} Li is deposited on Cu foils in the LiFSI-10DME electrolyte at a current density of 0.5 mA cm^{-2} .

Figure R21. AFM height images (a-c) and modulus distribution images (d-f) with different indentation depths. 1 mAh cm^{-2} Li is deposited on Cu foils in the LiFSI-1.4DME electrolyte at a current density of 0.5 mA cm^{-2} .

Figure R22. Average modulus of SEI with different indentation depths.

REVIEWERS' COMMENTS

Reviewer #1 (Remarks to the Author):

I am satisfied with the replies to my queries, and appreciate that several additional experiments were carried out to support the authors claims. I'm especially convinced by the additional experiments carried out at low current densities, which still show that that amorphous phase is present.

I now recommend that this manuscript is accepted for publication.

Reviewer #3 (Remarks to the Author):

The authors have thoroughly conducted additional experiments and simulations in response to the questions and concerns raised by the reviewer. With the new results and the explanations, the issues are clarified in the revision and the manuscript can be recommended for publication.

**Response to the Reviewers Comments on the Manuscript NCOMMS-
22-41990A**

Reviewer #1 (Remarks to the Author):

I am satisfied with the replies to my queries, and appreciate that several additional experiments were carried out to support the authors claims. I'm especially convinced by the additional experiments carried out at low current densities, which still show that that amorphous phase is present.

I now recommend that this manuscript is accepted for publication.

Response: We appreciate the reviewer for the highly positive comments and for recommending our manuscript for publication in Nature Communications.

Reviewer #3 (Remarks to the Author):

The authors have throughly conducted additional experiments and simulations in response to the questions and concerns raised by the reviewer. With the new results and the explanations, the issues are clarified in the revision and the manuscript can be recommended for publication.

Response: We sincerely thank the reviewer's very positive comments and for the acceptance of our work.